# Transcriptional repression of *NFKBIA* triggers constitutive IKK- and proteasome-independent p65/RelA activation in senescence

Marina Kolesnichenko[1,*] (ID), Nadine Mikuda[1], Uta E Höpken[2], Eva Kärgel[1], Bora Uyar[3] (ID),
Ahmet Bugra Tufan[1], Maja Milanovic[4] (ID), Wei Sun[5,†] (ID), Inge Krahn[1], Kolja Schleich[4], Linda von Hoff[1],
Michael Hinz[1], Michael Willenbrock[1], Sabine Jungmann[1], Altuna Akalin[3] (ID), Soyoung Lee[4],
Ruth Schmidt-Ullrich[1] (ID), Clemens A Schmitt[4] & Claus Scheidereit[1,**] (ID)

## Abstract

The IκB kinase (IKK)-NF-κB pathway is activated as part of the DNA damage response and controls both inflammation and resistance to apoptosis. How these distinct functions are achieved remained unknown. We demonstrate here that DNA double-strand breaks elicit two subsequent phases of NF-κB activation *in vivo* and *in vitro,* which are mechanistically and functionally distinct. RNA-sequencing reveals that the first-phase controls anti-apoptotic gene expression, while the second drives expression of senescence-associated secretory phenotype (SASP) genes. The rapidly activated first phase is driven by the ATM-PARP1-TRAF6-IKK cascade, which triggers proteasomal destruction of inhibitory IκBα, and is terminated through IκBα re-expression from the *NFKBIA* gene. The second phase, which is activated days later in senescent cells, is on the other hand independent of IKK and the proteasome. An altered phosphorylation status of NF-κB family member p65/RelA, in part mediated by GSK3β, results in transcriptional silencing of *NFKBIA* and IKK-independent, constitutive activation of NF-κB in senescence. Collectively, our study reveals a novel physiological mechanism of NF-κB activation with important implications for genotoxic cancer treatment.

**Keywords** DNA damage response; IκBα; NF-κB; SASP; senescence
**Subject Categories** Cell Cycle; Chromatin, Transcription & Genomics; Immunology
**The EMBO Journal (2021) 40: e104296**

## Introduction

Chemo- and radiotherapies activated oncogenes and shortened telomeres trigger via the DNA damage response (DDR) a terminal proliferative arrest called cellular senescence (Blagosklonny, 2014; Salama *et al*, 2014; Lasry & Ben-Neriah, 2015; Lee & Schmitt, 2019). The associated alterations include formation of senescence-associated heterochromatin foci (SAHF), increased synthesis of cell cycle inhibitors, including p21 (*CDKN1A*) and p16 (*CDKN2A*) and of inflammatory cytokines and chemokines that constitute the senescence-associated secretory phenotype (SASP) and a related, low-grade inflammation termed senescence inflammatory response (SIR) that affects surrounding tissues in a paracrine manner (Shelton *et al*, 1999; Lasry & Ben-Neriah, 2015). The epigenetically controlled cell cycle cessation serves as a cell-autonomous barrier to tumor formation (Braig *et al*, 2005; Collado *et al*, 2005; Reimann *et al*, 2010). Therefore, induction of senescence was considered as important in treating cancer and other pathologies. The inflammatory response, however, comprises factors that may instigate oncogenic transformation, cell migration, and cancer stemness (Acosta *et al*, 2008; Reimann *et al*, 2010; Chien *et al*, 2011; Freund *et al*, 2011; Jing *et al*, 2011; Acosta *et al*, 2013; Salama *et al*, 2014; Hoare *et al*, 2016; Milanovic *et al*, 2018).

The majority of SASP factors are transcriptional targets of NF-κB (Acosta *et al*, 2008; Kuilman *et al*, 2008; Chien *et al*, 2011; Freund *et al*, 2011; Jing *et al*, 2011; Lasry & Ben-Neriah, 2015). Although it is well established that NF-κB drives inflammatory gene expression in senescence, whether it also contributes to cell cycle arrest remained unclear. DNA double-strand breaks lead to rapid activation of NF-κB RelA/p65-p50, the most prevalent heterodimer (Smale, 2012). A signaling cascade that is activated by *ataxia*

---

1   Signal Transduction in Tumor Cells, Max Delbrück Center for Molecular Medicine, Berlin, Germany
2   Microenvironmental Regulation in Autoimmunity and Cancer, Max Delbrück Center for Molecular Medicine, Berlin, Germany
3   Bioinformatics/Mathematical Modeling Platform, Max Delbrück Center for Molecular Medicine, Berlin, Germany
4   Department of Hematology, Oncology, and Tumor Immunology, Charité-Universitätsmedizin, Berlin, Germany
5   Laboratory for Functional Genomics and Systems Biology, Max Delbrück Center for Molecular Medicine, Berlin, Germany
    *Corresponding author: Tel: +49 30 9406 3860; E-mail: marina.kolesnichenko@mdc-berlin.de
    **Corresponding author: Tel: +49 30 9406 3816; E-mail: scheidereit@mdc-berlin.de
    †Present address: Chemical Biology, UCSF, San Francisco, CA, USA

*telangiectasia mutated* (ATM) and poly(ADP-ribose) polymerase 1 (PARP1), and also depends on TNFR-associated factor 6 (TRAF6), converges on the IκB kinase (IKK) complex (Wu *et al*, 2006; Stillmann *et al*, 2009; Hinz *et al*, 2010). The latter is composed of the regulatory IKKγ subunit and the kinases IKKα and IKKβ (Hayden & Ghosh, 2012; Hinz & Scheidereit, 2014), which phosphorylate IκBα, targeting it for degradation by the 26S proteasome. Liberated NF-κB translocates to the nucleus and activates transcription of its target genes, including *NFKBIA*, encoding IκBα (Hinz *et al*, 2012). This negative feedback loop ensures that NF-κB activation is transient. Replication-, oncogene-, and therapy-induced senescence is associated with unresolved DNA damage and with constitutive NF-κB activation (Rodier *et al*, 2009; Chien *et al*, 2011; Freund *et al*, 2011; Jing *et al*, 2011).

Using cells of epithelial origin, both from transgenic mouse models and from human primary and cancer cell lines, we demonstrate here that DNA damage triggers two functionally distinct phases of NF-κB activation. Although the first, immediately activated phase is IKK- and proteasome-dependent and activates anti-apoptotic gene expression, the second NF-κB activation phase, occurring days later in senescence, is caused by a permanent silencing of *NFKBIA* transcription and is thus IKK- and proteasome-independent. We show that this second, IKK-independent phase of NF-κB is responsible for SASP, but not for the cell cycle arrest. Furthermore, SASP expression profile is generated *in vivo* and *ex vivo* in the absence of DNA damage solely by depletion of IκBα.

## Results

### DNA damage activates NF-κB in two distinct phases: a transient anti-apoptotic first and a persistent inflammatory second phase

To investigate the kinetics of activation of NF-κB, we first examined the establishment of senescence and SASP over time in human diploid fibroblasts (HDFs) and cancer cell lines that experienced DNA damage. Onset of senescence was marked by senescence-associated β-galactosidase (SA-β-gal) activity and elevated *CDKN1A* (p21) expression (Fig 1A and Appendix Fig S1A). Proliferation ceased 1–2 days following irradiation (IR), as expected, and cells entered a lasting senescent state (Appendix Fig S1B and C). Unresolved DNA damage, evidenced by γH2AX foci and phosphorylation of p53 at serine 15, was seen within minutes and hours and persisted through all time points. However, acute activation of upstream ATM and Checkpoint kinase 2 (Chk2) was observed predominantly only within minutes and hours after the IR treatment (Appendix Fig S1D and E). Strikingly, a single dose IR generated a biphasic NF-κB activation, with two temporally separate phases of its nuclear translocation and DNA binding, first, within hours and then days later (Fig 1B, and Appendix Fig S1D, F and G). An RNA-seq analysis revealed distinct transcriptomes in both NF-κB phases (Fig 1C and Tables EV1A–C). During the first phase, 326 transcripts showed significant upregulation (Dataset EV1A), which included direct targets of NF-κB, such as early response genes and negative feedback inhibitors of the pathway (*NFKBIA* and *TNFAIP3*; see Dataset EV1B). This transcript group was enriched for GO terms "cell cycle arrest" and "regulation of apoptotic process"

(Table EV1). In contrast, the 2,980 transcripts upregulated during the second phase were enriched for GO terms "inflammatory response", "immune response", and "response to wounding" (Table EV1). The biphasic NF-κB response thus coincided with two distinct transcriptomes, an anti-apoptotic first phase and a pro-inflammatory second phase.

We next analyzed murine tissues for IR-induced expression of the representative first- and second-phase target genes of NF-κB, *Nfkbia* and *Il6*, respectively. IR strongly activates NF-κB in various tissues (Li *et al*, 2001). Indeed, IR significantly induced *Nfkbia* mRNA in skin and kidney tissues only at early time points and of *Il6* mRNA only at late ones, representing the first and second NF-κB phases (Fig 1D). Other examples for induced phase-restricted gene expression in skin are *Tnfaip3* (only first phase) and *Ctsb* or *Cxcl3* (only second phase; Appendix Fig S1H).

To ensure that the expression of SASP during the second phase depended on NF-κB, we analyzed primary kidney cells from irradiated mice, which ubiquitously express the NF-κB super-repressor IκBαΔN (Krappmann *et al*, 1996; Schmidt-Ullrich *et al*, 2001). Compared to littermate controls, expression of *Il6* in the second phase was abolished, whereas *Cdkn1a* and *Cdkn2a* upregulation was unaffected (Appendix Fig S1I). These results further reveal that DNA damage activates NF-κB that drives SASP, but NF-κB is not essential for the proliferative arrest observed in senescence.

A strong response was seen in hair follicles (HF), which require NF-κB activation for development and morphogenesis (Schmidt-Ullrich *et al*, 2001). Upregulation of *Nfkbia* mRNA and IκBα protein was restricted to the first phase of NF-κB activation in HF following whole-body IR (Fig 1E and F). At Day 7 post-IR, IκBα expression in the proximal HF was strongly reduced, concomitant with an increase in IL-6 expression in the same region (Fig 1G and H). These results demonstrate that two distinct, subsequent NF-κB transcriptional programs also occur *in vivo*. Because IκBα is required to terminate NF-κB signaling, we postulated that loss of IκBα in senescence could trigger SASP.

### Loss of IκBα expression in senescence triggers the second phase of NF-κB activation and generates SASP

IκBα expression was either undetectable or strongly reduced in senescence in the different cancer- and non-transformed cell lines tested (Fig 2A and Appendix Fig S2A and B). Likewise, *NFKBIA* mRNA was upregulated only in the first phase (Fig 2A, right panel; Appendix Fig S2C), despite robust activation of NF-κB in both phases (Fig 1B). Repeat IR treatment, with the aim to re-stimulate first-phase-like transcriptional induction of *NFKBIA* in senescent cells, restored neither *NFKBIA* mRNA nor IκBα protein expression (Fig 2A left, lane 5, and right panel).

Activated oncogenes cause DNA strand breaks and induce DDR signaling, thereby promoting cellular senescence similar to cells exposed to DNA-damaging agents (Acosta *et al*, 2008; Coppe *et al*, 2008; Kuilman *et al*, 2008). Inducible activation of oncogenic RASV12 led to activation of NF-κB and expression of the representative SASP factor IL-8 (encoded by *CXCL8*) that negatively correlated with IκBα expression (Appendix Fig S2D and E). In summary, these data show that different pro-senescent triggers lead to loss of *NFKBIA* mRNA expression together with the onset of an NF-κB-driven SASP.

To investigate if experimental *NFKBIA* depletion mimics the second-phase NF-κB activation and SASP type gene expression, we knocked down *NFKBIA* (Fig 2B and Appendix Fig S2F). We

confirmed that DNA damage signaling was not altered due to *NFKBIA* knockdown (Appendix Fig S2F), and importantly, that constitutive NF-κB activation in untreated cells with *NFKBIA*

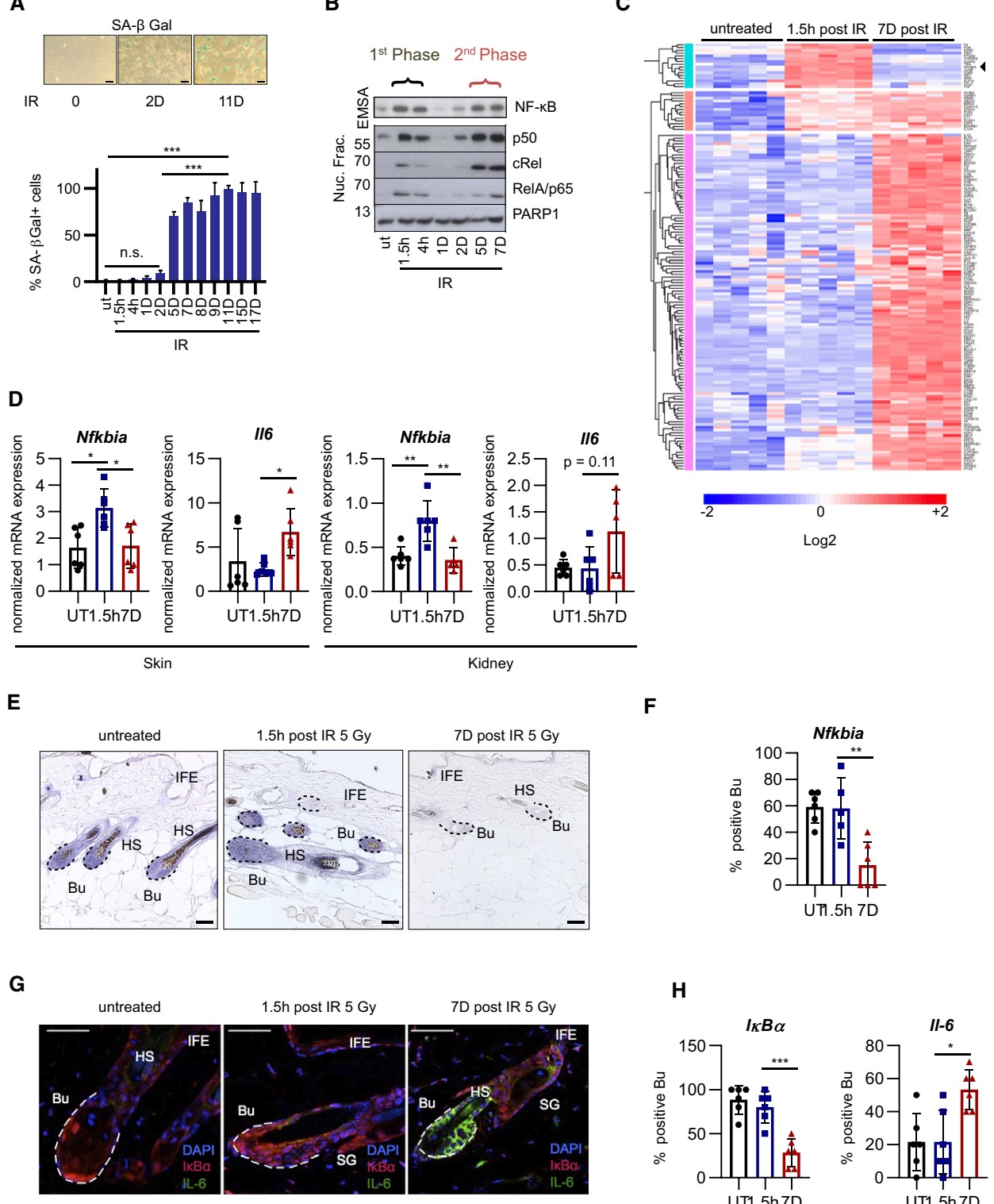

Figure 1.

**Figure 1.  NF-κB is activated in two distinct phases in response to DNA damage corresponding to anti-apoptotic and pro-inflammatory gene programs.**

A   SA-β-gal staining of U2-OS cells, harvested at indicated times post-irradiation (IR); h, hours; D, days; UT, untreated. $n = 3$. Scale bar: 50 µm. Bar graph: Significance was determined by Brown–Forsythe ANOVA with Dunnett's 3T test, using $n = 3$ biological replicates. SD shown. ***$P < 0.001$, n.s., not significant.

B   Nuclear fractions from U2-OS cells were harvested at indicated time points post-IR (20 Gy) as in (A). Top panel: NF-κB DNA binding was analyzed by EMSA. Lower panels: Western blot of indicated NF-κB subunits and PARP1 as loading control. The two phases of nuclear translocation and DNA binding are indicated. Representative gel shown from $n = 5$ biological replicates.

C   Gene expression analysis of two NF-κB phases following damage. RNA-seq analysis ($n = 5$ biological replicates per group) of U2-OS cells, either untreated or analyzed 1.5 h or 7 days after irradiation (20 Gy), as indicated. The heatmap shows significantly regulated genes ($\log_2$ value > 0.5 and $P$ value < 0.05). Black arrow points to *NFKBIA*. Significance determined by ANOVA with Bonferroni correction for multiple testing. See also Dataset EV1.

D   Female mice, 12–16 weeks old ($n = 6$ mice per condition (with three technical repeats per mouse), were irradiated (5 Gy) and sacrificed after 1 h (first phase), 7 days (second phase) or left untreated. Skin or renal tissue RNA was analyzed by RT–qPCR, shown as a mean ± SD. Significance was confirmed by ANOVA with Tukey multiple comparisons test. SD shown: *$P < 0.05$, **$P < 0.01$.

E   *In situ* hybridization using an anti-sense IκBα/*Nfkbia* riboprobe on longitudinal skin sections from female mice treated as in (D) for the time indicated. Representative sections shown from $n = 6$ mice per condition. Bu, bulge region (dashed lines), HS, hair shaft, IFE, interfollicular epidermis. Scale bar: 50 µm.

F   Quantitation of (E) from $n = 6$ mice per condition and 3–4 sections per mouse. Bulge regions positive for *Nfkbia* were counted and presented as percentage of total. One-way ANOVA analysis with Tukey multiple comparisons test was performed. SD shown. **$P < 0.01$.

G   Skin sections as in (E) analyzed by immunofluorescence with IκBα (red) or IL-6 antibody (green) and nuclear DAPI staining (blue). Dotted lines delineate hair follicles, Bu, bulge region, HS, hair shaft, IFE, interfollicular epidermis, SG, sebaceous gland. Scale bar: 50 µm.

H   Quantitation of (G) from $n = 6$ mice per condition and 3–4 sections per mouse. Bulge regions positive for IκBα or IL-6 were counted and presented as percentage of total. One-way ANOVA analysis with Tukey multiple comparisons test was performed. SD shown. *$P < 0.05$, ***$P < 0.001$.

Source data are available online for this figure.

knockdown (Fig 2B, lane 5) did not stem from a DDR (Appendix Fig S2F). These data indicate that knockdown of *NFKBIA*, irrespective of DNA damage, is sufficient for constitutive NF-κB activation.

We next asked which target genes would be regulated by constitutive NF-κB activation resulting from *NFKBIA* knockdown. Remarkably, the transcriptome of non-irradiated *NFKBIA*-depleted cells shared a strong overlap and the same GO terms with the senescent cells of the second phase (Fig 2C and D, and Table EV1). Thus, loss of IκBα in the absence of DNA damage was sufficient to activate transcription of SASP factors. IκBα rescue through ectopic overexpression of an *NFKBIA* mutant encoding the proteasome-insensitive IκBαS32AS36A blocked the induction of the *bona fide* SASP factors (Appendix Fig S2G), consistent with the conclusion that loss of IκBα drives expression of SASP. Secretion of a set of the identified cytokines and chemokines was confirmed by an antibody array (Appendix Fig S2H). Furthermore, supernatants from either senescent or IκBα-depleted cells induced migration of macrophages (Appendix Fig S2I).

Since we showed that suppression of *NFKBIA* was sufficient for SASP, we next asked whether it was also sufficient for senescence-associated proliferative arrest. At least in the case of Eμ-myc lymphomas, it has been suggested that NF-κB also positively regulates cell cycle arrest (Chien *et al*, 2011). We therefore analyzed proliferation of cells with *NFKBIA* knockdown. The duplication rate of cells with *NFKBIA* knockdown was comparable to that of control cells (Appendix Fig S3A). Similarly, BrdU incorporation in senescent cells with *NFKBIA* knockdown was not affected, despite constitutive activation of NF-κB (Appendix Fig S3B).

We next asked whether *in vivo*, constitutive NF-κB resulting from a knockout of *Nfkbia* would also trigger SASP, but not senescence-associated proliferative arrest. Since ubiquitous loss of IκBα causes early postnatal lethality (Beg *et al*, 1995; Klement *et al*, 1996), we generated intestinal epithelium-restricted (*villin-Cre* × floxed *Nfkbia*) knockout mice (Mikuda *et al*, 2020). Importantly, we observed hyperproliferation of cells, leading to crypt hyperplasia and enrichment of genes responsible for cell cycle progression. Increased proliferation (evidenced by increase in Ki67[+] cells) was observed both *in vivo* and in organoids derived from

these mice (Mikuda *et al*, 2020). Thus, constitutive NF-κB activation *in vivo* regulates proliferation and cell cycle progression in a positive manner. To determine if SASP factors could be detected *in vivo*, we performed GSEA analysis on epithelium from (*villin-Cre* × floxed) *Nfkbia* knockout mice and matched sibling controls (Fig 2E). Importantly, only enrichment of SASP-coding transcripts was detected, but not of those transcripts belonging to the broader category of cellular senescence, which in addition comprises factors involved in cell cycle regulation (Fig 2E and Appendix Fig S3C). We confirmed our findings by RT–qPCR by analyzing expression of selected SASP factors that are also targets of NF-κB, including *Ccl20*, *Icam1*, *Il6* and *Tnfa* (Fig 2F). These were significantly upregulated in the absence of DNA damage, evidenced by lack of p53 pSer15 nuclear foci (Appendix Fig S3D and E), confirming our conclusion in an *in vivo* setting.

### p65 regulates SASP in senescence but not proliferative arrest

Since we observed distinct NF-κB targets upregulated during the two phases (Dataset EV1B), we next asked which family member(s) were responsible for NF-κB activity in senescence. NF-κB showed robust DNA binding during both the first and the second phase (Fig 3A, lanes 7 and 13), consisting in each case of p65-containing heterodimers (lanes 10 and 16). We therefore knocked down *RELA*/p65 to analyze how it regulates the first and the second-phase transcriptomes. All NF-κB targets, including SASP-encoding genes, showed decrease in expression (Fig 3B) in cells bearing shRNA against *RELA*/p65 (Appendix Fig S4A). Nevertheless, knockdown of *RELA* did not rescue cells from senescence (Fig 3C and Appendix Fig S4A–C). These data confirm that p65 containing dimers regulate SASP but not the senescence-associated proliferative arrest.

### Suppression of *NFKBIA* in senescence through p65 pSer468

Since loss of IκBα in the second phase resulted from a decline of mRNA expression, we investigated its regulation by p65. The phosphorylation status of this subunit differentially determines

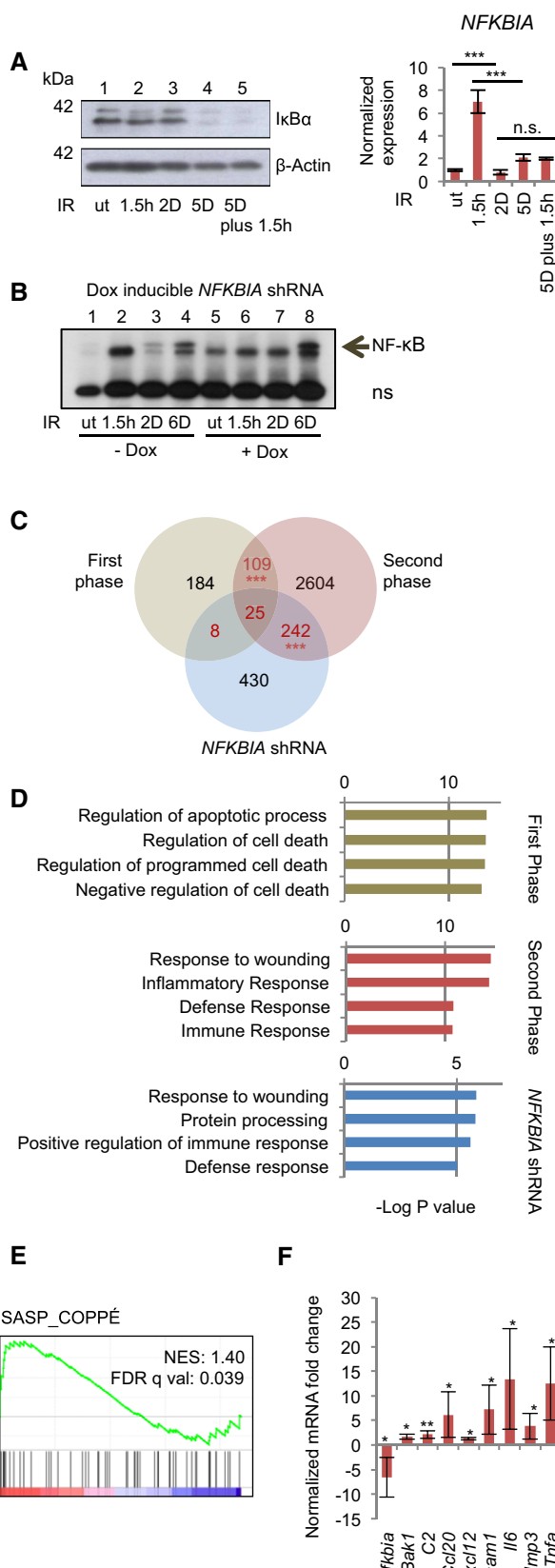

**Figure 2.**

Figure 2. Knockdown of *NFKBIA* in cell lines and knockout *in vivo* mimics second-phase NF-κB activity and triggers SASP.

A  Left panel: IκBα Western blot of U2-OS cells. Lane 1, untreated (ut); Lanes 2–4, single IR (20 Gy) at indicated time points prior to harvest; Lane 5, IR at 5 days prior to harvest, plus 1.5 h prior to harvest. Right panel: Samples treated as in the left panel, analyzed by RT–qPCR. Significance from *n* = 3 replicates per condition determined by ANOVA analysis with Tukey multiple comparisons test. SD shown. ***P* < 0.001.

B  *NFKBIA* knockdown by Dox-inducible shRNA in U2-OS cells. Cells were pre-treated with Dox to induce knockdown. For knockdown efficiency see Fig S2F. Cells were irradiated (20 Gy) or not, as indicated and described above. NF-κB activity was analyzed by EMSA. ns, non-specific band. A representative gel from *n* = 3 biological replicates is shown.

C  Venn diagram of RNA-seq analysis of genes activated following irradiation in the first and second NF-κB activity phases or following shRNA-mediated *NFKBIA* knockdown in non-irradiated cells (as in B). Numbers of overlapping genes are shown in red. Hypergeometric probability calculation showed significance between overlaps (adjusted with Bonferroni correction for multiple testing). SD shown. ***P* < 0.001.

D  Top GO terms obtained using DAVID Functional Annotation Bioinformatics Microarray 6.7 for first and second-phase genes and genes activated upon *NFKBIA* knockdown, as in (C), as indicated.

E  GSEA was performed comparing the IκBα $^{IEC-KO}$ gene list (*n* = 4 *villin-Cre* × *floxed* IκBα mice and *n* = 4 littermate controls (GSE 139251) (Mikuda *et al*, 2020), with a gene signature based on a list of SASP factors (Coppe *et al*, 2010). NES: normalized enrichment score. FDR: false discovery rate.

F  RT–qPCR analysis of indicated genes was performed using RNA extracted from the duodenum of 5- and 8-week-old control *villin-Cre* (*n* = 6) or *villin-Cre x floxed* IκBα mice (*n* = 6) both male and female. Expression is shown as fold change between littermates, paired *t*-test. SD shown. **P* < 0.05 ***P* < 0.01.

Source data are available online for this figure.

transcription of its target genes (Schmitz *et al*, 2004; Wietek & O'Neill, 2007). We therefore analyzed the phosphorylation status of p65 during the two phases (Fig 4A). Phosphorylation on p65 Ser536, the substrate site of IKK, peaked during the first phase and declined in the second. Unlike phosphorylation of p65 at S536 and S276, which enhance the p65 transactivation potential, phosphorylation at S468 is inhibitory and is mediated in part by GSK3β (Buss *et al*, 2004; Christian *et al*, 2016). Of note, GSK3β exhibits increased kinase activity in senescence to activate formation of SAHF, through downregulation of Wnt signaling (Ye *et al*, 2007). Indeed, nuclear phosphorylation on S468 increased in senescence (Fig 4A left and right panels). To determine if phosphorylation on S468 repressed expression of IκBα, we overexpressed p65 bearing a S468A mutation in cells where endogenous p65 was knocked down. Overexpression of p65 S468A rescued IκBα expression (Fig 4B). Similar results were observed with ectopic expression of wild-type p65, likely due to the abundance of the substrate in relation to the S468 kinase(s). As a negative control, we transfected the S276A mutant. Since phosphorylation at S276 is required for p65 activity (Christian *et al*, 2016), its overexpression did not rescue IκBα in senescence (Fig 4B).

This model strongly predicts that the *NFKBIA* locus is bound during the second but not the first phase by inhibitory p65, which is phosphorylated at serine 468. To explore this possibility, we performed chromatin immunoprecipitation (ChIP) with both p65 (total) and p65 pSer468 antibodies. Strikingly, unlike total p65,

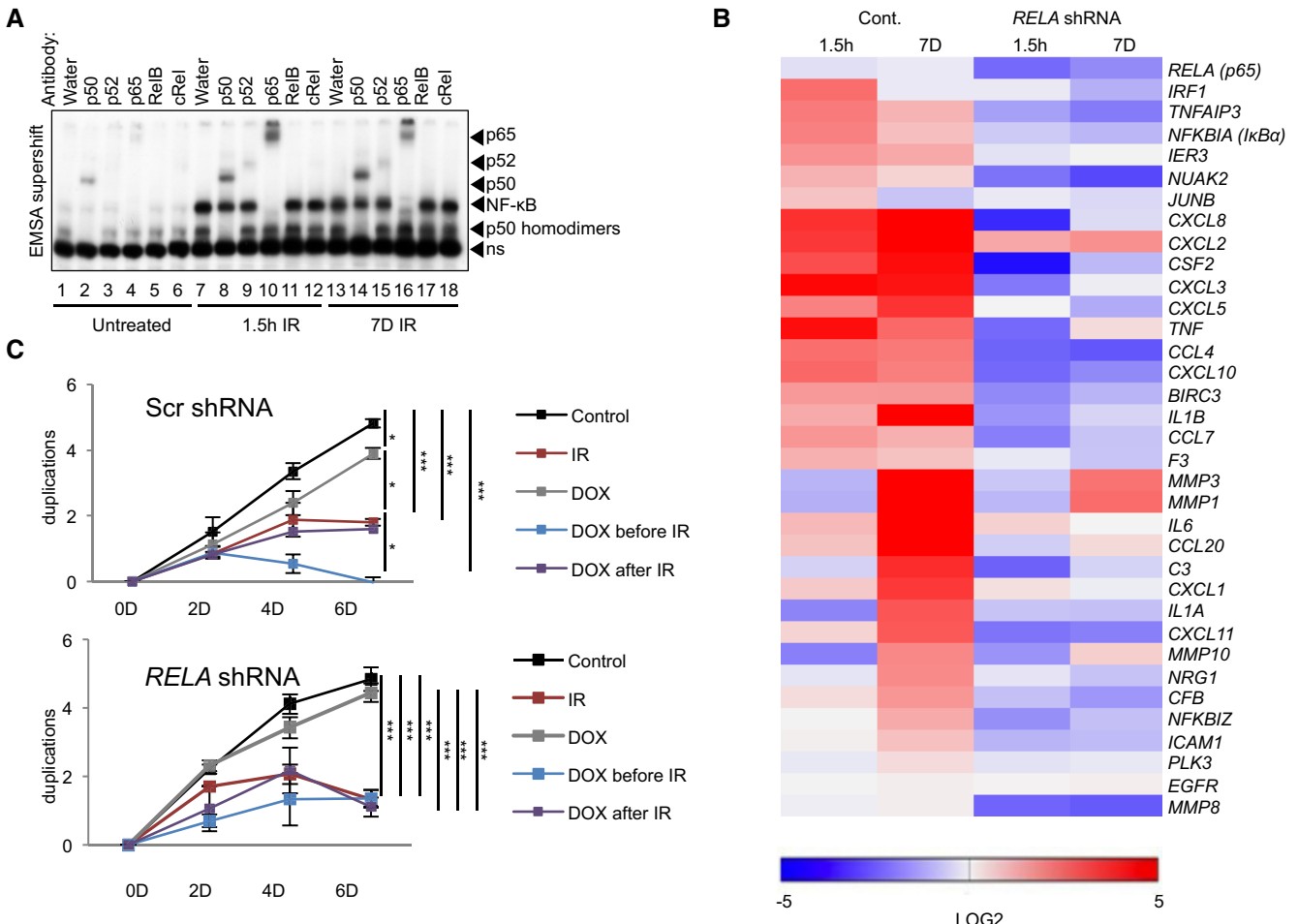

**Figure 3. RelA regulates SASP but not cell cycle arrest.**

A Nuclear fractions from either untreated or irradiated (20 Gy) U2-OS cells, 1.5 h or 7 days prior to harvest, were analyzed by EMSA with supershift analysis. For supershifting, lysates were incubated with antibodies or water, as indicated. Arrows point to shift location of a given subunit, due to antibody binding. ns, non-specific band. All antibodies were previously tested for supershift compatibility. Representative gel shown from n = 3 replicates.

B Expression of SASP targets of NF-κB obtained from RNA-seq analysis (Dataset EV1B), was quantitated using RT–qPCR from the shRNA (RELA) stably expressing U2-OS cells. For knockdown efficiency, see Appendix Fig S4A. Cells were treated with doxycycline (Dox) to induce knockdown of p65. Heatmap represents targets normalized to the untreated Scrambled control. Expression is shown as $\log_2$ change with P values < 0.05. Statistical analysis performed using ANOVA with Bonferroni correction for multiple testing. First-phase and second-phase samples were irradiated (20 Gy) 1.5 h and 7 Days prior to harvest, respectively. Analysis based on n = 3 biological replicates.

C Cell duplication was measured at time points indicated in U2-OS cells bearing either scrambled control or Dox-inducible shRNA against RELA (n = 3 biological replicates). Treatment with Dox was initiated at 2 days prior to IR (20 Gy) or after IR. Statistical significance in total duplication number between groups at day 6 was determined by ANOVA with Tukey multiple comparisons test. SD shown. *P < 0.05, ***P < 0.001.

Source data are available online for this figure.

which showed equivalent binding affinity during both phases, only p65 pSer468 showed strongly induced binding during the second phase (Fig 4C). Together, these data show that translational modifications on p65 lead to repression of its inhibitor-encoding gene, *NFKBIA,* in senescence.

To determine which kinases could contribute to the phosphorylation of S468, we analyzed the nuclear fraction of cells with a proteome kinase array. Of 45 selected kinases and proteins, 12 showed a significant change in response to IR (Fig 4D). As expected, factors associated with DNA damage, including p53 and Chk2, showed increased phosphorylation following irradiation, albeit with

differing kinetics (Fig 4D). Phosphorylation of GSK3α and β at the inhibitory sites, serine 21 and 9, respectively, was present in untreated cells and at 1.5 h post-IR, but was decreased in senescence, indicating that nuclear GSK was more active. This correlated with decreased abundance of β-catenin (Fig 4D), whose phosphorylation by GSK leads to its degradation (Kitagawa *et al*, 1999). Due to the previously described role of GSK in phosphorylation of p65 on serine 468, we knocked down GSK3β or inhibited GSK3β by lithium chloride and CHIR-99021 (Fig 4E–G). Both modes of interference diminished serine 468 phosphorylation and led to a partial restoration of IκBα expression (protein and mRNA) in senescence, with

concomitant decrease in expression of *IL6* (Fig 4G). These data show that changes in p65 phosphorylation contribute to an attenuation of *NFKBIA* expression in senescence.

### Second-phase NF-κB activation in senescence is independent of IKK signaling and the proteasome

We next investigated the contribution of the known regulators of the genotoxic stress-induced IKK pathway (Fig 5A) in the two NF-κB phases. The activation of ATM and IKK correlated only with the

first NF-κB phase and peaked in the first hours after IR followed by a decline afterward (Fig 5B). We also found that TRAF6 depletion only abrogated activation of the first, but not the second NF-κB phase (Fig 5C). Expression of target genes of NF-κB, *IL6* and *IL1A*, in the second phase was completely unaffected by depletion of ATM (Appendix Fig S5A and B). Furthermore, proteasomal inhibition abrogated IR-induced IκBα destruction in the first NF-κB phase, but not loss of IκBα in the second phase (Fig 5D). In line with this, upregulation of the *bona fide* SASP component IL-1α was not affected by proteasome inhibition (Fig 5D).

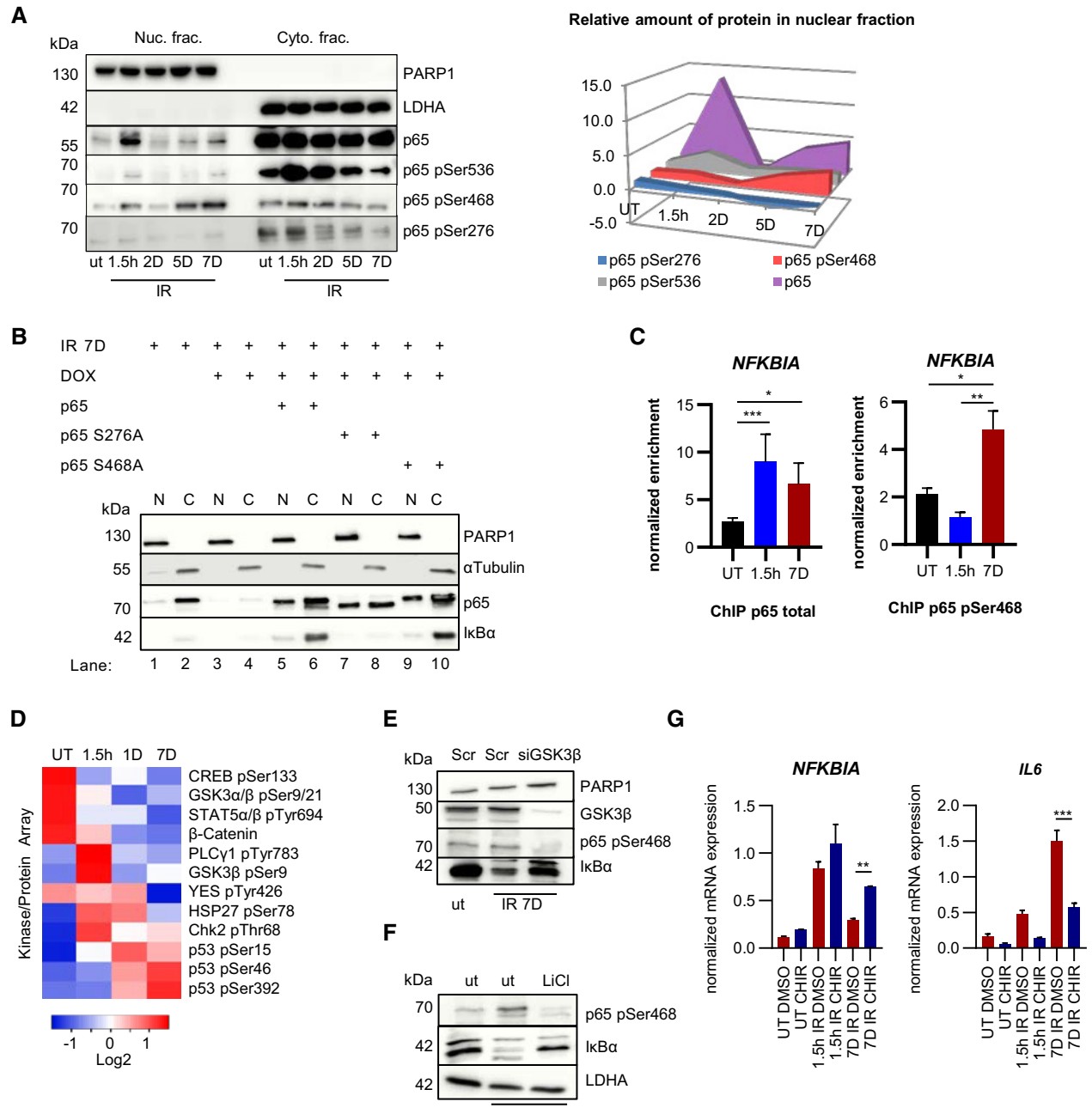

**Figure 4.**

◄

**Figure 4.**  p65 Ser 468 phosphorylation in senescence inhibits *NFKBIA* expression.

A  Nuclear (Nuc.) and cytoplasmic (Cyto.) fractions of U2-OS cells were analyzed at times indicated after IR exposure by SDS–PAGE Western blotting. PARP1 and LDHA are fractionation controls and phospho-specific p65 signals are indicated. Representative gels from *n* = 3 biological replicates are shown. Right panel: quantitation of nuclear p65 and phosphorylated p65 species (*n* = 3). Fold changes compared to untreated samples (ut) and time points post-IR are indicated on *Y* and *X* axes, respectively.

B  U2-OS cells treated with Dox to deplete endogenous p65 were irradiated (20 Gy) and transfected with plasmids encoding p65, p65-S276A or p65-S468A, as indicated. Nuclear (N) and cytoplasmic (C) lysates were analyzed by SDS–PAGE at day 7 post-IR. Representative gel from *n* = 3 biological replicates is shown. PARP1 and α-tubulin serve as fractionation and loading controls.

C  ChIP performed with U2-OS cells irradiated (20 Gy) 1.5 h or 7 days prior to assay, or left untreated (UT). Normalized relative enrichment of *NFKBIA* is shown (relative to input and two non-recruiting reference regions). Left panel: p65 antibody. *n* = 5 biological replicates with three technical repeats per biological replicate. Right: Antibody against p65 pSer468. *n* = 2 biological replicates with three technical repeats per biological replicate. Statistical significance was determined by ANOVA with Tukey multiple comparisons test. SD shown. *P < 0.05, **P < 0.01, ***P < 0.001.

D  Kinase proteome array was performed with nuclear fractions of U2-OS cells that were either left untreated (UT) or irradiated (20 Gy) at time points indicated prior to harvest. Kinase binding to membrane was quantitated (*n* = 2) and only significant results (*P* < 0.05) are shown. Significance was determined by ANOVA with Tukey multiple comparisons test.

E  U2-OS cells left untreated (ut) or irradiated (20 Gy) 7 days before harvesting, as indicated. Two days prior to harvest cells were transfected with siRNA against GSK3β or scrambled control (Scr). Whole cell lysates were analyzed by SDS–PAGE and Western blotting. A representative example from *n* = 3 experiments is shown.

F  U2-OS cells were left untreated or exposed overnight to 10 mM LiCl, with or without prior irradiation, as indicated, and analyzed as in (E). Representative data from *n* = 3 experiments are shown.

G  RT–qPCR of U2-OS cells irradiated (20 Gy) either 1.5 h or 7 days prior to harvest. CHIR denotes treatment with GSK3 inhibitor CHIR-99021 (10 nM overnight). DMSO overnight treatment was used as control. Quantitation was performed and statistical significance obtained from *n* = 3 replicates, using ANOVA with Tukey multiple comparisons test. SD shown. *P < 0.05, **P < 0.01, ***P < 0.001.

Source data are available online for this figure.

We eliminated IKKβ using CRISPR/Cas9 (Fig 5E). Loss of IKKβ impeded only first-phase NF-κB activation (Fig 5E, lanes 2 and 6). Remarkably, IKKβ was not required for the second phase of NF-κB activation (Fig 5E, lanes 4 and 8).

Our conclusion was further corroborated by identification of genes that depended on p65 expression (Fig 3B), yet showed unaltered expression in senescent cells with CRISPR mediated knockout of *IKBKB* (Fig 5F). These IKK-independent genes encoded many *bona fide* SASP factors, such as IL8 and metalloproteinases MMP1 and 3 (Fig 5F, lower half of the panel). Likewise, siRNA-mediated IKKγ depletion only diminished p65 nuclear translocation at 1.5 h, but not 5 days following IR (Appendix Fig S5C). To determine if *CHUK*/IKKα that regulates non-canonical signaling might regulate SASP, we checked expression of select NF-κB targets following *CHUK* knockdown (Appendix Fig S5D). No significant downregulation of most SASP genes was detected.

All these data substantiate that the second phase of NF-κB activation is independent of the IKK cascade induced by DNA damage and independent of proteasomal destruction of IκBα.

## Discussion

Most NF-κB activation pathways depend on IKKs (Hoffmann & Baltimore, 2006; Hinz & Scheidereit, 2014). Here we provide a physiologically relevant context for IKK-independent activation of NF-κB both in human cells lines and in murine models *in vivo*. We found that in DNA damage-induced senescence of epithelial cells, two interconnected events comprising a decline in IKK phosphorylation and a drop in transcription of the inhibitor of the pathway, *NFKBIA* (IκBα), initiate a persistent, IKK-independent activation of NF-κB (Fig 6).

We and others have previously shown that irreparable DNA damage leads to cellular senescence and to SASP driven by NF-κB (Rodier *et al*, 2009; Chien *et al*, 2011; Jing *et al*, 2011). However, since DNA damage triggers prompt activation of NF-κB, whose

immediate transcriptional targets do not feature SASP, it was not clear how NF-κB could be responsible for two such drastically different transcriptomes arising from the same initial stimulus. Here we demonstrated both *in vivo* and *ex vivo* that a single dose of DNA damage sequentially activates two temporally and functionally distinct transcriptomes of NF-κB, separated by a span of several days. An anti-apoptotic first phase is driven by an ATM-, PARP-1-, and TRAF6-dependent IKK signaling cascade (Stilmann *et al*, 2009; Hinz *et al*, 2010), resulting in proteasomal destruction of IκBα. A pro-inflammatory second phase occurs in senescence and comprises SASP (Fig 6). Importantly, we demonstrate that the second phase of NF-κB and expression of the majority of the SASP genes are both IKK- and proteasome-independent. A fraction of transcripts that were IKK-dependent in senescence could be regulated in an alternative manner that does not require phosphorylation in the activation loop of IKKβ. Indeed, we have recently shown that basal activity of IKKβ suffices for its interaction with EDC4 (Enhancer of Decapping 4) and for post-transcriptional stabilization and destabilization of scores of transcripts, including of *CXCL8* and *TNFA* (Mikuda *et al*, 2018). It is therefore possible that IKK-dependent regulation of some SASP genes occurs at the level of their RNA stability. It is also possible that additional phosphorylation sites on IKK, not analyzed in this paper, contribute to its activation and to regulation of IKK-dependent genes expressed in senescence.

Interestingly, absence of IKK at the instant of DNA damage abolishes only the first phase of NF-κB activation, but does not affect the second phase. This suggests that the first phase is not required for the second, and that the changes accumulated over time activate distinct signaling pathways that enable the second phase of NF-κB. In accordance with this, we show that post-translational modifications on p65 contribute to the two distinct transcriptomes. IKK phosphorylates p65 on serine 536 within minutes following DNA damage, however in senescence a switch in phosphorylation from serine 536 to 468, and to recruitment of p65-pSer468 to the *NFKBIA* locus, leads to repression of *NFKBIA*. GSK3β, which is hyperactive in senescence (Ye *et al*, 2007), phosphorylates p65 at Ser 468.

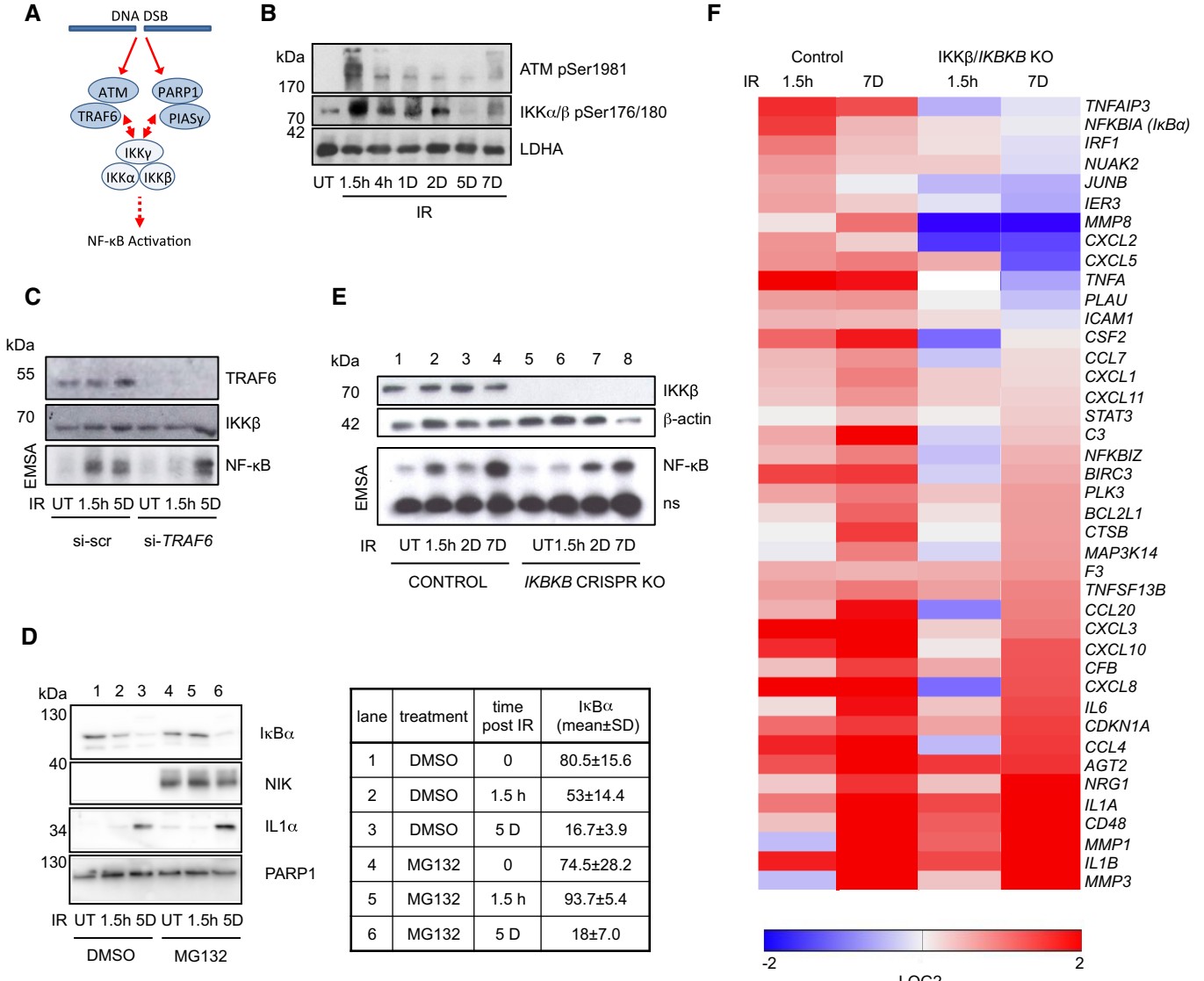

**Figure 5. The second phase is IKK- and proteasome-independent.**

A  Diagram showing DNA double-strand break (DSB) induced first-phase NF-κB activation through an ATM/TRAF6- and PARP1/PIASy-dependent IKK activation mechanism (Wu *et al*, 2006; Stilmann *et al*, 2009; Hinz *et al*, 2010).

B  SDS–PAGE Western blot of whole cell lysates of U2-OS cells irradiated (20 Gy) at time points indicated prior to harvest. ut, untreated. LDHA. Representative gel shown from *n* = 4 biological replicates.

C  U2-OS cells transfected with *TRAF6* siRNA or scrambled control siRNA and TRAF6 and IKKβ levels analyzed by Western blotting (top panels). NF-κB activity was determined by EMSA in untreated cells (ut) and after IR at indicated time points (lower panel). ns, unspecific band. Representative gels are shown from *n* = 2 biological replicates.

D  U2-OS cells were exposed to IR at indicated time points prior to harvest and analyzed by Western blotting for levels of IκBα, NIK, IL-1α, and PARP1. Treatment with proteasomal inhibitor MG132 (10 μM) started 4 h prior to harvest. NIK serves as positive control for efficient proteasome inhibition. Quantitation from *n* = 4 biological replicates and SD is shown for IκBα (right panel). Brown–Forsythe ANOVA test: *P* = 0.0005.

E  U2-OS CRISPR *IKBKB* knockout and control cell lines were irradiated (20 Gy) and harvested at indicated time points. NF-κB activity was analyzed by EMSA (lower panel). ns indicates an non-specific band. Upper panel, Western blot analysis of actin and IKKβ. Representative gels shown from *n* = 3 replicates.

F  Irradiated U2-OS control and *IKBKB* knockout cells (as above) were analyzed by RT–qPCR at the time points indicated for expression of NF-κB target genes identified by RNA-seq (Dataset EV1). Significantly regulated targets, cutoff set at *P* = 0.05 (as determined by ANOVA with Tukey multiple comparison), are shown.

Source data are available online for this figure.

However, inhibition of GSK3β did not completely reinstate IκBα expression, indicating that additional kinases and/or epigenetic changes may contribute to downmodulation of *NFKBIA* expression in senescence.

Silencing of *NFKBIA* in senescence results in IKK-independent and persistent activation of NF-κB (Fig 6). Consequently, the second phase can be mimicked without induction of DNA damage by inactivation of the *NFKBIA* gene, as we have shown in human cell

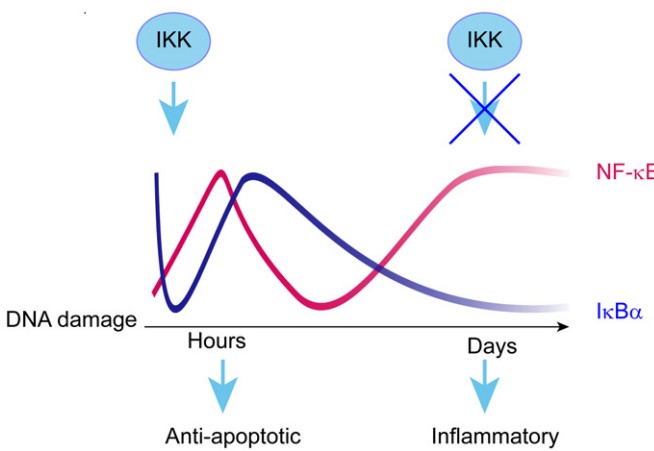

**Figure 6.  Schematic model of the two-phase NF-κB activation after DNA damage.**

DNA damage triggers two phases of NF-κB activation and two distinct transcriptomes: anti-apoptotic and inflammatory. NF-κB is rapidly activated through the known genotoxic stress-induced IKK cascade, resulting in proteasome-dependent IκBα degradation and expression of first-phase target genes, including IκBα. NF-κB activation in senescence is caused by a loss of IκBα expression through a largely IKK- and proteasome-independent mechanism and drives SASP expression.

cultures and the murine knockout model. Nonetheless, the gene expression repertoire of *NFKBIA* knockout or knockdown cell lines does not encompass the entire senescent inflammatory response likely because full-featured senescence relies on activation of additional regulators including TORC1, MAPK, Toll like receptors, Notch1/TGF-β, and C/EBPβ, and also on epigenetic changes that firmly establish proliferative arrest (Narita *et al*, 2011; Kolesnichenko *et al*, 2012; Serrano, 2012; Herranz *et al*, 2015; Hoare *et al*, 2016).

It was previously suggested that in addition to promoting SASP, p65 reinforces cell cycle arrest in senescent Eμ-myc lymphomas expressing the anti-apoptotic protein Bcl-2 (Chien *et al*, 2011). However, Bcl-2 was also shown to attenuate cell cycle progression independently of its anti-apoptotic functions (Zinkel *et al*, 2006). Our findings clearly demonstrate that although NF-κB negatively regulates expression of several cell cycle genes, knockdown of p65 does not disturb the established cell cycle arrest in epithelial cells and murine model described here (Dataset EV1A–C). Furthermore, we did not observe establishment of senescence either in human cell lines bearing knockdown or knockout of *NFKBIA*, or in the murine mouse model. We therefore conclude that NF-κB mediates only the inflammatory phenotype, but not the cell cycle arrest in senescence.

NF-κB is constitutively activated in a variety of cancer types (Ben-Neriah & Karin, 2011). Deletion of the *NFKBIA* gene or low expression of IκBα protein serves as a major mechanism of permanent NF-κB activation in a non-classical form of glioblastoma, where it is associated with poor prognosis in patients and resistance to proteasome and IKK inhibitors in clinical trials (Idbaih *et al*, 2011; Takeuchi & Nawashiro, 2011; Kinker *et al*, 2016; Raizer *et al*, 2016). Importantly, classical SASP factors and NF-κB targets, such as IL-8, IL-6, and metalloproteinases are constitutively expressed in these tumors, where they are suggested to fuel tumor growth and

invasion (Puliyappadamba *et al*, 2014). Our results imply that in the milieu of established senescence and concomitant loss of IκBα, inhibition of the IKK-signalosome or the proteasome would be ineffective in suppressing SASP in tumor therapies. On a positive note, direct inhibition of NF-κB would inhibit detrimental SASP, while leaving the cell cycle arrest intact.

# Materials and Methods

### Transfection/transduction

pTRIPz clones IκBα, p65, IKKβ, and IKKγ and Scrambled or pGIPZ-IκBα and pGIPZ-scrambled (Dharmacon, Lafayette, USA) were transfected into HEK293T cells and supernatant used for transduction as described in manufacturer's protocol (http://dharmacon.ge lifesciences.com/uploadedfiles/resources/ptripz-inducible-lentiviral-manual.pdf). Clonal selection was performed using puromycin. Doxycycline hydrochloride was added daily (2 μg/ml, Sigma). Unless specified otherwise, Dox treatment was done for 5–6 days prior to harvest. As controls, cells inducibly expressing scrambled shRNAs were treated with Dox.

CRISPR knockout cells were generated as described previously (Mikuda *et al*, 2018).

### Preparation of murine tissues from *in vivo* experiments

All mouse protocols in this study followed the regulatory standards of the governmental review board (Landesamt Berlin), Reg. G007/08, G-0029/15, G 0082/13, G0358/13, and X9013/11). B6;129P2-Nfkbia^tm1Kbp and Tg(Vil-cre)20Syr) mice were sacrificed at 5 or 8 weeks of age. Additionally, 12- to 16-week-old C57Bl6/N female mice were sacrificed either 1.5 h or 7 days post-whole-body IR (5Gy).

### Nuclear cytoplasmic fractionation, Western Blot analysis, and EMSA

Nuclear cytoplasmic fractionation, Western Blot analysis, and EMSA performed as described previously (Mikuda *et al*, 2018).

### In situ hybridization

*In situ* hybridization was performed as described previously (Schmidt-Ullrich *et al*, 2001).

### Antibody array

Proteome profiler (R&D Systems) antibody array was performed on 1 ml of culture medium according to manufacturer's protocol. Quantitation was performed with FusionCapt Advanced software.

### Kinase array

(R&D Systems) was performed on nuclear lysates according to manufacturer's protocol. Quantitation was performed with FusionCapt Advanced software.

### Quantitative RT–PCR

Quantitative RT–PCR (RT–qPCR) was performed using the CFX96 Real Time System (Bio-Rad) and GoTaq® qPCR Master Mix (Promega), using a minimum of two reference genes (*TBP, Rpl13a, HRTP1*).

### RNA-seq

RNA samples were prepared in quintuplicates and extracted using Trizol reagent according to manufacturer's instructions (Thermo Fisher). Stranded mRNA sequencing libraries were prepared with 500 ng total RNA according to manufacturer's protocol (Illumina). The libraries were sequenced in $1 \times 100 + 7$ manner on HiSeq 2000 platform (Illumina).

### Antibodies

For a list of antibodies, See Table EV2.

## Data availability

The RNA-seq data from this publication have been deposited to the GEO database (www.ncbi.nlm.nih.gov/geo) and assigned the identifier GSE158743 (http://www.ncbi.nlm.nih.gov/geo/query/acc.cgi?acc=GSE158743).

**Expanded View** for this article is available online.

## Acknowledgements

We would like to thank Dr. Asanka Gunawardana (Institute of Biometry and Clinical Epidemiology, Charité Berlin) for help with statistical analysis, Dr. Hans-Peter Rahn and Kerstin Krueger for help with FACS, Wei Chen and the MDC sequencing facility for RNA-seq analysis and Sarina Hilke, Manasa Reddy Gummi, Nadine Burbach for excellent assistance. This work was supported in part by BMBF, CancerSys project ProSiTu, Deutsche Forschungsgemeinschaft DFG SCHE277/8-1 and Helmholtz Association iMed to CS and from ProSiTu and the German Cancer Aid (Deutsche Krebshilfe) grant number 110678 to CAS. Open Access funding enabled and organized by ProjektDEAL.

## Author contributions

Conceptualization: MK, CS; Data curation: WS; Formal analysis: MK, BU, EK, UEH; Funding acquisition: CS; Investigation: MK, NM, EK, MH, ABT, MM; Project administration: MK; Resources: CAS, CS, RS-U, SL, AA; Supervision: CS; Validation: IK, MW, SJ, LH, KS; Visualization: MK, NM, EK, CS, Writing original draft: MK; Writing—review & editing: CS, MK, CAS, RSU, UEH, EK.

## Conflict of interest

The authors declare that they have no conflict of interest.

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
