## [Review Process File · The EMBO Journal]

Transcriptional repression of NFKBIA triggers constitutive IKK- and proteasome-independent p65/RelA activation in senescence

Marina Kolesnichenko, Nadine Mikuda, Uta Höpken, Eva Kaergel, Bora Uyar, Ahmet Tufan, Maja Milanovic, Wei Sun, Inge Krahn, Kolja Schleich, Linda von Hoff, Michael Hinz, Michael Willenbrock, Sabine Jungmann, Altuna Akalin, Soyong Lee, Ruth Schmidt-Ullrich, Clemens Schmitt, and Claus Scheidereit

DOI: [10.15252/embj.2019104296](https://doi.org/10.15252/embj.2019104296)

Corresponding author(s): *Claus Scheidereit* (scheidereit@mdc-berlin.de), *Marina Kolesnichenko* (marina.kolesnichenko@mdc-berlin.de)

Review Timeline:

Submission Date:	18th Dec 19
Editorial Decision:	7th Feb 20
Revision Received:	27th Oct 20
Editorial Decision:	1st Dec 20
Revision Received:	7th Dec 20
Accepted:	9th Dec 20

Editor: Hartmut Vodermaier

Transaction Report:

Thank you for submitting your manuscript on DNA damage-dependent NF- κ B activation and senescence induction to The EMBO Journal. I apologize for the long time it has taken to get back to you with the outcome of its review, which was due to limited referee availability at the end of the year, and delayed referee reports. We have now received a complete set of comments from three expert referees, copied below for your information.

As you will see from these comments, all referees acknowledge the potential interest of your new findings and conclusions. However, they clearly also remain unconvinced that the presented data provide sufficiently decisive evidence to strongly support such conclusions. The most salient concerns are related to insufficient statistical analyses and biological replicates, including mouse work controls in agreement with field standards. Moreover, the referees ask for better senescence phenotype characterization, and more convincing evidence for the proposed NF- κ B activation mechanism in the second wave, esp. re p65 phosphorylation and IKK α involvement. I will not go through all individual points of criticism in detail here, given their clear description in the three reports, but have to conclude here that the study is currently not a good candidate for an EMBO Journal article, at least not in its present form.

That said, given the overall potential interest and the fact that the issues raised may in principle be addressable, I would nevertheless like to give you an opportunity to respond to the referees' comments by way of a revised version of the study. I realize that this would clearly require significant further time and efforts, and given the unclear outcome of such major revision work hope you appreciate that I am not in a position to make strong predictions on the outcome of an eventual re-evaluation; I would therefore also understand if you were to rather seek rapid publication with minor revisions in an alternative venue. Should you decide to attempt revision for The EMBO Journal, please note that our single-revision-round policy would make it important to comprehensively and diligently answer/address all points raised at this stage. To facilitate this, I would be happy to discuss an extension of the revision period, during which publication of any competing work elsewhere would not have a negative impact on our final assessment of your own study. Also, please do not hesitate to contact me already during the early stages of revision in case you should have any questions regarding this decision or the reviewer comments and how to address them.

REFEREE REPORTS

Referee #1:

This MS represents a potentially important contribution to our understanding of NF κ B activation and hence SASP of senescent cells. Although there is consensus that NF κ B activation is the major driver of SASP in senescent cells, there is surprisingly little known re. the mechanism of activation of SASP. This MS attempts to fill that important gap. The authors start by showing that DNA damage causes an early and late biphasic activation of NF κ B - the early phase primarily responsible for expression of anti-apoptotic genes and the later phase for activation of SASP. Mechanistically, the authors propose that "the second phase is activated days later in senescent cells but is independent of IKK and the proteasome. An altered phosphorylation status of p65, in part driven by GSK3 β , results in transcriptional silencing of NFKBIA and IKK independent, constitutive activation of NF κ B in senescence". While the authors have focused on an important problem and I think have made important insights, the mechanistic analyses are deficient. The following points should be addressed prior to publication.

Major points.

1. In the abstract, the authors state "An altered phosphorylation status of p65, in part driven by GSK3 β , results in transcriptional silencing of NFKBIA and IKK independent, constitutive activation of NF- κ B in senescence". The data in support of this idea appear to be 3 fold 1) that p65S468 phos increases in senescence; 2) that ectopic expression of non-phospho p65S468A abolishes repression of I κ Ba; 3) that inhibition of GSK3b blocks phosphorylation of p65S468 and increases expression of I κ Ba. The 2nd of these is essentially invalidated because WT p65 has the same activity - thus there is nothing to implicate regulation by phosphorylation in this result. There also seems to be a fundamental paradox in this model - that p65pS468 represses expression of I κ Ba but not other SASP genes, such as IL6. The authors model strongly predicts a difference in activity between WT and p65pS468 which is not shown, and presumably that p65pS468 binds to repressed I κ Ba but not expressed IL6 genes.

2. Figure 5D is a key figure but is not convincing. The quality of the blot is poor, so it is difficult to make quantitative assessments. The dynamic range of the assay (i.e. difference in most of the band intensities) is low. The conclusion that MG132 and bortezomib affect I κ Ba in the 1st phase but not the 2nd requires quantitation from replicates.

Minor points.

1. Figure 1D. The authors should test expression of additional 1st and 2nd phase target genes in these tissues.

2. Figure 1G. Representative images are shown. Is it possible to quantitate data from multiple tissue

sections?

3. Figure S2D shows data from transformed HEK293 and HCT116 cells. The relevance of these cells is questionable since they do not senesce, presumably. Particularly in HEK293 cells, there is no apparent selective repression of I κ B α at the 5 day timepoint. I suggest these data are removed from the MS.

4. The authors state "Inducible activation of oncogenic RASV12 led to activation of NF- κ B and expression of the representative SASP factor IL-8 (encoded by CXCL8) that negatively correlated with I κ B α expression (Appendix Fig S2F-G)." This is an overstatement of the data because I κ B α is lowest at 2D when IL8 is not expressed. This also argues against downregulation of I κ B α being sufficient for NF κ B activation.

5. Figure 2B. Knock down of I κ B α should be confirmed by western blot. Why is Nf κ B observed as a doublet in some but not all cases?

6. Figure 2C. The authors show overlap of the 2nd phase SASP and genes activated by I κ B α knock down. While the overlap appears substantial and significant, the authors should calculate the fold enrichment and p value of the overlap compared to a predicted random overlap. How many shRNAs have been used to show that knock down of I κ B α induces senescence? Senescence is fundamentally a stress response - therefore off target effects of shRNAs could generate this result. Is there a correlation across several shRNAs between knock down of I κ B α and activation of senescence.

7. Figure 2E. Are these increases in gene expression significant?

8. Figure 4D. There are much more specific inhibitors of GSK3 β than LiCl, I think.

9. Figure S4B. Confirm knock down of ATM by western blot.

Referee #2:

In this article, Scheiderei and colleagues demonstrated that DNA damage triggers two phases of NF- κ B activation, which are both functionally and mechanistically different from one another and are separated by a span of several days. In the first phase which occurs immediately following DNA damage, NF- κ B drives the expression of anti-apoptotic genes. This first phase is driven by an ATM-, PARP-1- and TRAF6-dependent IKK signaling cascade which triggers proteasomal destruction of I κ B α and is terminated through I κ B α (NFKBIA) re-expression. The second NF- κ B activation phase occurs several days after senescence have been established and comprises of expression of different pro-inflammatory SASP components. NF- κ B activation in this phase is caused by a permanent silencing of NFKBIA transcription, and thus it is both IKK- and proteasome-independent. This interesting study uncovers novel mechanism of regulation of expression of pro-inflammatory cytokines following DNA damage. The authors present some very interesting tissue culture based work on the role of NF- κ B in promoting a distinct signature of gene expression in DNA damage

induced senescent cells and the importance of NFKBIA in the cascade. The in-vivo work that examine the role of NFKBIA in the two phases on NF- κ B activation is also interesting, but needs to be further substantiated. Unfortunately, there are several major issues with this study as outlined below. I hope the comments would help the authors to improve the study and in any case they have to be addressed before publication.

General Striking Drawbacks:

1. Lack of proper statistical analysis: Some of the graphs representing experiments are either not supplemented with statistical analysis, or are supplemented with statistical analysis shown on the panel, but are not referred to in the legend of the same figure.

The authors should perform the statistical analysis for all their experiments, before they draw any conclusions and present the analyses according to the accepted standards. The kind of analysis done in each experiment and the amount of repeats should be stated clearly in the figure legend.

Notable absence of statistical analysis present on the following figures:

- Fig 1A: Between the "ut", "2d" and "11d" groups.
- Fig 2E: Between the control villin-Cre to the villin-Cre x floxed I κ B α groups in each of the genes tested.
- Fig 3B: Between all the groups shown in the graphs at time-points "2D" and "6D".
- Fig S1A: Between the "ut", "2d" and "11d" groups.
- Fig S1B: Between "Untreated" and "IR 10Gy" groups, at time-points "2D" and "6D".
- Fig S2E: Between "ut", "1.5h" and "5D" groups in all the cell lines that were examined.
- Fig S2G-H: Between "ut" and "1.5h Tam." groups, "ut" and "5D Tam." groups and between "1.5h Tam." and "5D Tam." groups.
- Fig S2H: Between "ut", "1.5h" and "6D" groups, "ut" and "ut dox" groups and between "ut dox" and "6D dox".
- Fig S2J:
 - NFKBIA: Between "empty vec." to "I κ B α 3236" groups and between the "I κ B α 3236" and "I κ B α 3236 IR" groups (show to statistical significance)
 - IL1a, CXCL8 and CCL20: Between "empty vec." to "empty vec. IR" and between "empty vec. IR" to "I κ B α 3236 IR".
- Fig S4A: Between "ut" and "90min" groups, "ut" and "7D" groups, and between "90min" and "7D" groups.
- Fig S4B: Between "SCR ut", "SCR 1.5h" and "SCR 7D" groups, "ATM si ut", "ATM si 1.5h" and "ATM si 7D" groups and between "SCR 1.5h" to "ATM si 1.5h" groups (in all graphs).

Notable absence of the type of the statistical analysis present in the figure legends as follows:

- Fig 1C, Fig 1E, Fig 2A, Fig 52E, Fig S3B

2. Lack of statement regarding the number of biological repeats: For each presented experiment, the number of individual repeats should be stated in the figure legend!

The following figures lack this essential information:

- Fig 1: 1B
- Fig 2: 2A, 2B
- Fig 3: 3A, 3B
- Fig 4: 4A, 4B, 3C, 4D
- Fig 5: 5B, 5C, 5D, 5E

- Fig S1: S1C, S1D, S1E
- Fig S2: all
- Fig S4: all

Most strikingly, in two of the in-vivo experiments the number of mice used was 2 (in figures 1E, 2A and S3A). This amount of repeats is insufficient to conclude a biological significance and are below of any standard. No conclusions can be drawn from 2 mice!!!

3. Lack of information about magnification: In some of the figures showing IF or IHC staining, a scale bar is missing, specifically, in figures 1A and S3B. Furthermore, in all the figures showing staining of cells/tissues, namely, figures 1A, 1F, 1G, S1C and S3B, there is no mention of the magnification and scale used.

Main comments:

1. The characterization of senescence in U2-OS cells is not well established.

The authors have checked for the expression of the senescence markers SA- β -Gal and p21 (CDKN1A) in U2-OS cells (Fig S1A). This is not sufficient to prove that the cells are senescent. The senescence state of the cells should be further characterized by checking for more markers of senescence. For example, the expression levels of CDK inhibitors such as p15 (CDKN2B), p16 (CDKN2A) and p53 can be evaluated in both protein and RNA levels. It is not clear at all if U2-OS cells are arrested. In order to verify the cell cycle arrest in the cells, BrdU incorporation in U2-OS cells can be performed.

2. The mouse tissues were not studied in sufficient detail to allow to reach the conclusions authors suggest

a. In Fig 1D murine tissues were irradiated and analyzed for expression of NFKBIA and IL-6 in both skin and kidney tissues. The authors mention in the text a strong activation of NF- κ B, however the results are not shown. This data is critical for verification of the results shown.

b. The in-vivo system should be characterized in terms of senescence and activation of DNA damage. Relevant markers could be genes that are part of the ds-DDR pathway, for example, p-ATM, p-CHK2, p-p53 and γ H2AX.

c. The accumulation of senescent cells in the in-vivo system can be done by checking for senescence markers such as p15, p16 and p21 in both protein and RNA levels. The SA- β -Gal staining will also be helpful for this purpose.

d. Importantly, the statistical test used for analyzed the data is incorrect, for the authors compare between 3 groups overall and are using t-test. The more appropriate statistical text for this kind of results is to compare all 3 groups using one-way ANOVA test.

e. Tissues of I κ B α Δ N mice (Fig 1E) require characterization of DDR, senescence and first phase markers, as suggested above for wt mice.

f. Similarly, tissues of villin-Cre x floxed mice were not studied to the sufficient details that allow to reach the drawn conclusions. The characterization of these samples as suggested above could help to resolve this shortcoming.

3. In Fig 3 knockdown should be verified and senescence should be better characterized:

In Fig3A: This figure shows the RNA-seq analysis of the expression levels of SASP targets of NF- κ B

following knocking down of p65. The authors should complement this results with a western blot showing the knockdown of p65 in all the relevant experimental groups.

In Fig3B: This figure shows cell proliferation following knockdown of p65 and irradiation. Cell proliferation by itself is not sufficient to conclude about the senescence state of the cells. The authors should check for different senescence markers (as suggested above in 1). Furthermore, cell proliferation could be complemented with BrdU incorporation assay, to show cell division.

4. Fig S1C: Quantification of the staining of both p65 and γ H2AX in all the groups would help to straighten the conclusion.

In order to evaluate the DDR in the cells, the authors could also check for more markers of the ds-DDR pathway, for example, p-ATM, p-CHK2, p-p53 and γ H2AX.

5. Fig S2A-D: For both BJ and WI-38 cells the authors show in S1D and S1E respectively the levels of p65 following irradiation in different time points. However, in S2B and S2D, the authors use HEK293 and HCT cells respectively to measure $\text{I}\kappa\text{B}\alpha$ levels, but they do not show the changes in the levels of p65 to verify the functionality of this systems. Therefore, the authors should include those results.

Minor comments:

1. In the paper the authors use various primers from both human and mouse origin in order to check the changes in expression of different genes. However, the authors do not list the primer sequences that were used for the various experiments. Please add all the relevant sequences to the "materials and methods" section.

2. The sex of the mice used in the experiments should be mentioned in the text and figure legend. Did the authors performed the experiments on both female and male mice? Is there a significant difference between the sexes?

3. Fig S1C: Quantification of the staining of both p65 and γ H2AX in all the groups would help to straighten the conclusion.

In order to evaluate the DDR in the cells, the authors could also check for more markers of the ds-DDR pathway, for example, p-ATM, p-CHK2, p-p53 and γ H2AX.

4. Fig S2A-D: For both BJ and WI-38 cells the authors show in S1D and S1E respectively the levels of p65 following irradiation in different time points. However, in S2B and S2D, the authors use HEK293 and HCT cells respectively to measure $\text{I}\kappa\text{B}\alpha$ levels, but they do not show the changes in the levels of p65 to verify the functionality of this systems. Therefore, the authors should include those results.

5. In Fig 1E the expression levels of IL-6 seems to be significantly lower in Δ N mice, in "1.5h" and "6D" time points, compared to the " Δ N untreated" mice. Furthermore, the expression levels of p16 seems to be significantly higher in the " Δ N 6D" group. Can the authors explain these two phenomena?

6. In Fig 2B the authors performed a knockdown $\text{I}\kappa\text{B}\alpha$ and check for the changes in NF- κ B over time. The efficiency of the knockdown should be evaluated and presented.

7. Fig 4C-D: Figure legends are displaced and appear in the legend of figure 5.

8. FigS1B: The amount of irradiation presented on the figure legend (10 Gy) and on the graph itself (20 Gy) are different.

9. FigS1C: In the figure legend the authors indicate that white arrows point to high nuclear p65 levels. Such arrows are missing from the figure itself.

10. Fig S2A-D: In each cell line the authors use different housekeeping gene for normalization. This genes are not usually used as housekeeping genes. The authors should try to normalize the western blots using one or two classical housekeeping genes, for example, b-Actin/GAPDH.

11. The authors performed a knockdown ATM and check for the changes in IL-6 and IL-1 α over time - Fig S4B. The efficiency of the knockdown should be evaluated.

12. The levels of the housekeeping gene used in the "nuclear" part of the figure S4C are not constant between the different lanes and seems to be changing in the same trend as p65. The authors should consider to choose a different housekeeping gene, or to repeat the experiment.

Referee #3:

The manuscript by Kolesnichenko and colleagues describes a novel mechanism of NF- κ B activation following DNA damage. Their data shows that a delayed second wave of NF- κ B activation promotes a senescence associated secretome characterised by cytokines and chemokines. The authors demonstrate that expression of NFKB1A which encodes for the NF- κ B inhibitor I κ B is lost following DNA damage and facilitates a subsequent delayed NF- κ B activation wave. The authors propose that the phosphorylation of the RelA subunit at S468 by GSK3 β promotes the transcriptional silencing of NFKB1A following DNA damage. Moreover, the authors propose that the second wave of NF- κ B activity leading to the senescence associated secretory phenotype is independent of the proteasome and IKK kinases. The data presented is of high quality and includes the use of a number of genetic models, in vitro and in vivo. Overall the authors conclusions are justified but the proposed mechanism leading to the regulation of NF- κ B following DNA damage is not sufficiently supported by the data presented.

Major points

1. Although the authors have focused on the RelA subunit of NF- κ B in this study, the data shown in Figure 1 B would indicate that the cRel subunit is significantly increased during the second wave of activation compared to the first. In addition the EMSA data shown in figure 2b suggests that the DNA binding complexes may be different in the second wave. Have the authors looked at the potential role for cRel or other subunits? If not can they speculate on the potential relevance of these data?

2. The author demonstrate a lack of NFKB1A expression following the DNA damage event. What are the expression patterns for NFKB1B (I κ B β) and NFKB1E (I κ B ϵ)? Are they detected in the RNA-seq data and do they follow a similar pattern to that of I κ B α ? Can the authors rule out a role for these related proteins?

3. The data in Figure 3 indicates that RelA promotes the expression of the SASP gene set but does not promote cell cycle arrest. The authors should measure the expression of known regulators of senescence associate cell cycle arrest such as CDKN1A and CDKN2A in control and RelA knockdown cells to demonstrate whether the lack of effect on cell cycle arrest may be related to the expression of such factors.

4. The data presented on the role of phosphorylation of p65 at S 468 is not very convincing and doesn't strongly support the authors proposal that phosphorylation at S468 controls of NFKB1A expression. The western blot data shown in figure 4 B should be quantified and I κ B α expression normalised to levels of overexpressed p65 to more clearly demonstrate that the S468A mutation increases the expression of I κ B α . The authors should also probe these samples with an antibody for pS468 and S267. Similarly the data shown in figure 4C and 4D should also be quantified and the level of I κ B α normalised to the levels of p65 pS468. For these two experiments the level of total p65 should also be measured. The analysis of the effects of 10mM LiCl treatment on p65 phosphorylation needs to be controlled by cells treated with an equivalent concentration of NaCl or similar salt.

5. The authors use proteasome inhibitors and deletion of IKK β and NEMO to show that the second wave of NF- κ B activity is independent of IKK and the proteasome. However these data do not rule out the possible involvement of IKK α and do not justify the authors conclusion. The authors need to rule out the possible role of IKK α in order to come to the conclusion that the activity is IKK-independent.

Minor points:

1. The RNA-seq data should be made- available in open access database such as EMBL ArrayExpress or NCBI GEO as per the journal guidelines.
2. For most experiments shown the figure legends do not indicate how many independent replicate experiments were performed.
3. The figure legend for figure 4 parts C and D appears on the legend for figure 5.

Our responses to the three referees:

Referee #1:

This MS represents a potentially important contribution to our understanding of NFkB activation and hence SASP of senescent cells. Although there is consensus that NFkB activation is the major driver of SASP in senescent cells, there is surprisingly little known re. the mechanism of activation of SASP. This MS attempts to fill that important gap. The authors start by showing that DNA damage causes an early and late biphasic activation of NFkB - the early phase primarily responsible for expression of anti-apoptotic genes and the later phase for activation of SASP. Mechanistically, the authors propose that "the second phase is activated days later in senescent cells but is independent of IKK and the proteasome. An altered phosphorylation status of p65, in part driven by GSK3β, results in transcriptional silencing of NFKBIA and IKK independent, constitutive activation of NFkB in senescence". While the authors have focused on an important problem and I think have made important insights, the mechanistic analyses are deficient. The following points should be addressed prior to publication.

We thank the referee for this insightful appraisal and address the points raised by performing suggested experiments.

Major points.

1. In the abstract, the authors state "An altered phosphorylation status of p65, in part driven by GSK3 β , results in transcriptional silencing of NFKBIA and IKK independent, constitutive activation of NF- κ B in senescence". The data in support of this idea appear to be 3 fold 1) that p65S468 phos increases in senescence; 2) that ectopic expression of non-phospho p65S468A abolishes repression of I κ B α ; 3) that inhibition of GSK3b blocks phosphorylation of p65S468 and increases expression of I κ B α . The 2nd of these is essentially invalidated because WT p65 has the same activity - thus there is nothing to implicate regulation by phosphorylation in this result. There also seems to be a fundamental paradox in this model - that p65pS468 represses expression of I κ B α but not other SASP genes, such as IL6. The authors model strongly predicts a difference in activity between WT and p65pS468 which is not shown, and presumably that p65pS468 binds to repressed I κ B α but not expressed IL6 genes.

We are grateful to the referee for pointing out these very important points and would like to clarify how all three support our model. "1) that p65S468 phosphorylation increases in senescence". We have demonstrated this using nuclear/cytoplasmic fractionation, followed by SDS-PAGE western blot analysis (Fig 4A). In addition, we have now performed chromatin immunoprecipitation with both p65 (total) and p65 pSer468 antibodies (new Fig 4C). These new data now confirm that p65 pSer468 is inducibly recruited to the NFKBIA gene during the second phase but not during the first phase of NF- κ B activation. "2) that ectopic expression of non-phospho p65S468A abolishes repression of I κ B α ;" We thank the referee for bringing up the point regarding the phospho-mutants. Compared to endogenous p65, the transfected p65 variants are expressed at significantly higher levels in the nuclear fractions (in Fig 4B, compare lanes 1 with 5, 7 and 9). At this high level, the endogenous kinase is presumably unable to phosphorylate a sufficiently high molar fraction of p65 at S468, which would be required to silence NFKBIA transcription. We think this is the reason why excess ectopic wildtype p65 rescues I κ B α expression, out-titrating endogenous p65 with its high fraction of S468 phosphorylation. The intact Serine 276 phosphorylation site in the in the S468A mutant, in contrast, can be sufficiently phosphorylated to drive NFKBIA transcription and indeed this serine is functionally essential: The ectopic p65 S276A is unable to rescue transcription. For referee inspection, we add a new experiment in Fig R1, which demonstrates strong Ser468 phosphorylation only of nuclear endogenous p65 (lane 2), but not of ectopic serine mutants or ectopic wildtype p65 (lanes 4, 6 and 8).

We are also grateful to the referee to bringing up the other important point that "The authors model strongly predicts a difference in activity between WT and p65pS468 which is not shown" Our ChIP data confirm this for NFKBIA. p65pS468 is inducibly recruited to the NFKBIA gene only during the second phase of NF- κ B activation (New Fig 4C). As the referee suggested, the phosphorylation state of p65 at the various residues likely influences the transcription of NF- κ B target genes in a differential and promoter-context-specific manner. In addition to the ChIP data for NFKBIA, we now also point this out in the manuscript with additional references: "The phosphorylation status of this subunit (p65) differentially determines transcription of its target genes" (Schmitz et al., 2004; Wietek & O'Neill, 2007)". Different effects on target genes by individual p65 phosphorylation sites were also shown previously for a limited number of phosphorylation sites (analyses did not include S468) by Hochrainer K., et al., J. Biol. Chem. 2013. Conceptually, phosphorylation of individual sites can lead to the recruitment of co-activators or repressors, which in turn depends on additional promoter/enhancer-selective interactions.

2. Figure 5D is a key figure but is not convincing. The quality of the blot is poor, so it is difficult to make quantitative assessments. The dynamic range of the assay (i.e. difference in most of the band intensities) is low. The conclusion that MG132 and bortezomib affect I κ B α in the 1st phase but not the 2nd requires quantitation from replicates.

We have now repeated the MG132 experiment (shown as new Fig 5D) and quantitated the results from 4 biological replicates. The quantitation further confirms our previous findings that proteasomal degradation accounts for loss of I κ B α protein in the first, but not the second phase of NF- κ B activation. The quantitation is shown as right panel of new Fig 5D.

Minor points.

1. *Figure 1D. The authors should test expression of additional 1st and 2nd phase target genes in these tissues.*

We have repeated the experiment using a total of 18 mice (with n = 6 for each experimental group), and additionally tested other first and second phase target genes of NF- κ B (new Fig 1D and Appendix Fig S1H). We have quantitated the data and show that also *in vivo*, *Nfkb1a* is inducibly expressed both in the skin and the kidney during the first phase, but not the second phase. And on the contrary, SASP factors including *Il6*, *Cxcl3* and *Ctsb* are preferentially expressed during the second phase but not during the first phase (Fig 1D and Appendix Fig S1H). These findings confirm and extend our previous conclusion that a single dose of DNA damage through NF- κ B drives two different and temporally separated sets of genes *in vivo*.

2. *Figure 1G. Representative images are shown. Is it possible to quantitate data from multiple tissue sections?*

Yes, we have now repeated the experiment using a total of 18 mice, with n = 6 mice per group and have quantitated the data. We have changed the panels in order to show longitudinal sections of hair follicles that better depict the stem-cell compartment. We confirm that a significant difference in expression of *Nfkb1a* exists between the first and the second phase of NF- κ B activation, also on mRNA level *in vivo* (new Fig 1E). The quantitation for new panel 1E is shown as new panel 1F and of panel G is shown as new 1H. All these findings further support our conclusion that during the second phase suppression of I κ B α does not occur through proteasomal degradation, but rather due to suppression of mRNA production. This allows for lasting suppression of the inhibitor and for constitutive activation of NF- κ B.

3. *Figure S2D shows data from transformed HEK293 and HCT116 cells. The relevance of these cells is questionable since they do not senesce, presumably. Particularly in HEK293 cells, there is no apparent selective repression of I κ B α at the 5 day timepoint. I suggest these data are removed from the MS.*

In these cell types, there is likewise a strong decrease of I κ B α protein levels at 5 days after irradiation compared to untreated cells and a biphasic activation of NF- κ B. However, since we have not addressed these cell types in further detail in the article, we followed the referee's suggestion and have removed the two panels.

4. *The authors state "Inducible activation of oncogenic RASV12 led to activation of NF- κ B and expression of the representative SASP factor IL-8 (encoded by CXCL8) that negatively correlated with I κ B α expression (Appendix Fig S2F-G)." This is an overstatement of the data because I κ B α is lowest at 2D when IL8 is not expressed. This also argues against downregulation of I κ B α being sufficient for NF κ B activation.*

We thank the referee for raising this point. In brief, suppression of *Nfkb1a* alone is sufficient to activate NF- κ B, as several earlier cell culture studies have shown, and as we and others have demonstrated in mouse models (Beg, A.A. et al., 1995, Genes Dev.; Klement, J.F., 1996, Mol. Cell. Biol.; Mikuda, N. et al. 2020, J. Pathol.). However, increased accumulation of

SASP (proteins) at later time points could be due to accumulation of their transcripts overtime. Furthermore, recent studies showed that mRNAs of SASP factors are stabilized in senescence via the p38 pathway and AUF1 (AU rich factor 1) (reviewed in Faget DV et al, 2019, Nat Rev Cancer). It is also known that numerous inflammatory factor-encoding transcripts contain AU rich elements in their 3' UTR, which can lead to mRNA destabilization, as we and others have shown (Hao, S. & Baltimore, D., 2013, PNAS; Mikuda, N. et al., 2018, EMBO J). Whether NF- κ B or IKK influences stability of mRNA in senescence and if this affects the timing of SASP expression is an interesting question, but is beyond the scope of the current study. Therefore, there may be several reasons why IL-8 protein expression is not immediately observable after the induction of NF- κ B.

5. Figure 2B. Knock down of I κ B α should be confirmed by western blot. Why is Nf κ B observed as a doublet in some but not all cases?

As suggested, we have now included a blot showing the Dox-inducible knockdown of *NFKBIA* (new Appendix Fig S2F). We believe that the dimer of the two NF- κ B bands which is observed in the EMSA in Fig 2B at late time points results from different composition and migration of the heterodimers. We see it only when using whole cell lysates for EMSA analysis (Fig 2B). We determined that p65 is the dominant subunit and is responsible for NF- κ B binding both, during the first and the second phase (new Fig 3A, performed with nuclear extracts). We also analyzed other subunits that contain transcription activation domain (c-Rel and RelB) by EMSA/supershifting and found that they did not contribute to nuclear DNA-binding complexes in senescence. p65 was the essential subunit for gene activation during the first and the second phase (new Fig 3A and 3B).

6. Figure 2C. The authors show overlap of the 2nd phase SASP and genes activated by I κ B α knock down. While the overlap appears substantial and significant, the authors should calculate the fold enrichment and p value of the overlap compared to a predicted random overlap. How many shRNAs have been used to show that knock down of I κ B α induces senescence? Senescence is fundamentally a stress response - therefore off target effects of shRNAs could generate this result. Is there a correlation across several shRNAs between knock down of I κ B α and activation of senescence.

We have now analyzed the significance level between overlaps in the three groups of genes and have included the data into the main figure (new Fig 2C). The analysis confirms our previous observation that the overlaps e.g. between the second phase (IR 7 days) and *NFKBIA* shRNA (untreated) are significant. In contrast, the overlap between the first phase (IR 1.5h) and *NFKBIA* shRNA (untreated) is not. Please see new Fig 2C (and the legend) for statistical analysis and data.

We have used two siRNAs and shRNA clones for *NFKBIA* depletion and additionally performed a CRISPR/Cas knockout of *NFKBIA* in human primary cells lines to show that its knockdown or knockout leads to transcription of SASP factors. We have also used a mouse model to confirm our findings *in vivo*, that depletion of *Nfkb1a* leads to constitutive NF- κ B activation and SASP transcription (new Fig 2E and F). We would like to emphasize that knockdown of *NFKBIA* does not result in senescence (or proliferative arrest), but only in expression of SASP. This is a key point, because we can show in the different cell types that constitutive NF- κ B generated by I κ B α downregulation drives transcription of SASP genes, but not proliferative arrest.

7. Figure 2E. Are these increases in gene expression significant?

The expression data now includes standard deviation and not standard error of mean and the data are significant for all genes shown. We apologize for the omission in the previous version. We have now included p values for the analyses as part of the figure and legend (new Fig 2F). In addition, please see statistical analyses, which take variance and normality into account (EMBOJ-2019-104296_ForRefereesInspection_Statistics).

8. *Figure 4D. There are much more specific inhibitors of GSK3b than LiCl, I think.*

Yes, and we have now additionally used CHIR99021 to inhibit GSK, and confirmed our previous observations. We include these data as a new Fig 4G. The data show that treatment with CHIR leads to an increase in *NFKBIA* expression in the second phase. Expression of *IL6*, in contrast, is decreased with CHIR (new Fig 4G). We neither reach complete restoration of *NFKBIA*, nor complete suppression of *IL6*, likely due to additional signaling pathways regulating both *NFKBIA* and *IL6* expression in senescence. Nonetheless, we show that suppression of GSK during the second phase of NF- κ B activation can in part rescue I κ B α expression.

9. *Figure S4B. Confirm knock down of ATM by western blot.*

As requested by the referee, we have now included a western blot that shows the efficient depletion of the ATM protein after its siRNA mediated knockdown (new Appendix Fig S5A).

Referee #2:

In this article, Scheidereit and colleagues demonstrated that DNA damage triggers two phases of NF- κ B activation, which are both functionally and mechanistically different from one another and are separated by a span of several days. In the first phase which occurs immediately following DNA damage, NF- κ B drives the expression of anti-apoptotic genes. This first phase is driven by an ATM-, PARP-1- and TRAF6-dependent IKK signaling cascade which triggers proteasomal destruction of I κ B α and is terminated through I κ B α (NFKBIA) re-expression. The second NF- κ B activation phase occurs several days after senescence have been established and comprises of expression of different pro-inflammatory SASP components. NF- κ B activation in this phase is caused by a permanent silencing of NFKBIA transcription, and thus it is both IKK- and proteasome-independent. This interesting study uncovers novel mechanism of regulation of expression of pro-inflammatory cytokines following DNA damage. The authors present some very interesting tissue culture based work on the role of NF- κ B in promoting a distinct signature of gene expression in DNA damage induced senescent cells and the importance of NFKBIA in the cascade. The in-vivo work that examine the role of NFKBIA in the two phases on NF- κ B activation is also interesting, but needs to be further substantiated. Unfortunately, there are several major issues with this study as outlined below. I hope the comments would help the authors to improve the study and in any case they have to be addressed before publication.

We thank the referee for the appraisal, and for the very helpful suggestions, critiques and insights. We could further substantiate our findings by expanded *in vivo* work and extensive statistical analysis, as recommended.

General Striking Drawbacks:

1. *Lack of proper statistical analysis: Some of the graphs representing experiments are either not supplemented with statistical analysis, or are supplemented with statistical analysis shown on the panel, but are not referred to in the legend of the same figure.*

The authors should perform the statistical analysis for all their experiments, before they draw any conclusions and present the analyses according to the accepted standards. The kind of analysis done in each experiment and the amount of repeats should be stated clearly in the figure legend.

We apologize for this omission and have now performed additional statistical analyses and modified the figures and legends correspondingly. For ease of overview, we have summarized the analyses performed in the table file EMBOJ-2019-104296_ForRefereesInspection_Statistics for referees' inspection. We are happy to report that statistical analyses further validated our observations. We thank the referee for this important suggestion.

Notable absence of statistical analysis present on the following figures:

- Fig 1A: Between the "ut", "2d" and "11d" groups.
- Fig 2E: Between the control villin-Cre to the villin-Cre x floxed I κ B α groups in each of the genes tested.
- Fig 3B: Between all the groups shown in the graphs at time-points "2D" and "6D".
- Fig S1A: Between the "ut", "2d" and "11d" groups.
- Fig S1B: Between "Untreated" and "IR 10Gy" groups, at time-points "2D" and "6D".
- Fig S2E: Between "ut", "1.5h" and "5D" groups in all the cell lines that were examined.
- Fig S2G-H: Between "ut" and "1.5h Tam." groups, "ut" and "5D Tam." groups and between "1.5h Tam." and "5D Tam." groups.
- Fig S2H: Between "ut", "1.5h" and "6D" groups, "ut" and "ut dox" groups and between "ut dox" and "6D dox".
- Fig S2J:
 - NFKBIA: Between "empty vec." to "I κ B α 3236" groups and between the "I κ B α 3236" and "I κ B α 3236 IR" groups (show to statistical significance)
 - IL1a, CXCL8 and CCL20: Between "empty vec." to "empty vec. IR" and between "empty vec. IR" to "I κ B α 3236 IR".
- Fig S4A: Between "ut" and "90min" groups, "ut" and "7D" groups, and between "90min" and "7D" groups.
- Fig S4B: Between "SCR ut", "SCR 1.5h" and "SCR 7D" groups, "ATM si ut", "ATM si 1.5h" and "ATM si 7D" groups and between "SCR 1.5h" to "ATM si 1.5h" groups (in all graphs).

We have included statistical analyses for the figures listed above as well as for the new figures and indicate now the type of analysis and replicate numbers in the figure legends. Please see EMBOJ-2019-104296_ForRefereesInspection_Statistics for summary, which, however, does not include a description of statistical analyses performed on RNAseq data, which are detailed in the Methods section in Supplemental Information.

In summary, our statistical analyses confirm our observations. As the referee suggested, we performed one-way ANOVA (with Tukey multiple comparison test (MCT)). We tested homoscedasticity with Brown-Forsythe ANOVA (and show Dunnett's multiple comparison) and we also tested for normal distribution. In case the distribution was not normal, we additionally performed a non-parametric ANOVA (Kruskal-Wallis test, supplemented with Dunn's multiple comparison test).

Please note that statistical tests for S2H and S4A are not listed in the table because they have been replaced by a western blot in Appendix Fig S2F and ChIP (new Fig 4C, for statistics see entry for Fig 4C in Table R1).

Notable absence of the type of the statistical analysis present in the figure legends as follows:

- Fig 1C, Fig 1E, Fig 2A, Fig 52E, Fig S3B

We apologize for the incomplete presentation and have now amended our legends to include the required information.

2. Lack of statement regarding the number of biological repeats: For each presented experiment, the number of individual repeats should be stated in the figure legend!

The following figures lack this essential information:

- Fig 1: 1B
- Fig 2: 2A, 2B
- Fig 3: 3A, 3B
- Fig 4: 4A, 4B, 3C, 4D
- Fig 5: 5B, 5C, 5D, 5E
- Fig S1: S1C, S1D, S1E
- Fig S2: all
- Fig S4: all

We have added the missing information to the figure legends of the figures named above and included the information in the legends of the new figures.

Most strikingly, in two of the in-vivo experiments the number of mice used was 2 (in figures 1E, 2A and S3A). This amount of repeats is insufficient to conclude a biological significance and are below of any standard. No conclusions can be drawn from 2 mice!!!

We apologize for this shortcoming and have now repeated animal experiments with 18 mice, n = 6 mice per group. We have exchanged panels in Fig 1E (new Fig 2E and Appendix Fig S11). Fig 2E includes longitudinal sections that better depict the bulb region of the hair follicle. In addition, we have quantitated hair bulbs positive for *Nfkb* staining by counting several sections from each mouse. We then used ANOVA and additionally compared three conditions to each other. Statistical analysis confirms our previous observation that unlike in the first phase of NF- κ B activation, where *Nfkb* is expressed, in the second phase the expression is lost at the mRNA level.

For new Appendix Fig S11, we have performed experiments using cells from four mice per condition. It is important to point out that the four pairs were not littermate controls this time, due to lack of availability of appropriate mice. Therefore, we believe that the variation observed (we are showing standard deviation throughout the manuscript and not SEM) could be due to variation between litters.

3. Lack of information about magnification: In some of the figures showing IF or IHC staining, a scale bar is missing, specifically, in figures 1A and S3B. Furthermore, in all the figures showing staining of cells/tissues, namely, figures 1A, 1F, 1G, S1C and S3B, there is no mention of the magnification and scale used.

We have included scale bars in a corner of each image panel. In addition, the magnification and the type of confocal microscope used is provided in the Supplemental Materials and Methods section.

Main comments:

1. The characterization of senescence in U2-OS cells is not well established.

In order to better characterize senescence in U2-OS cells, we have additionally performed BrdU and FACS analyses and included these data as a new Appendix Fig S1C. We have furthermore analyzed the expression of *CDKN2A* and provide an additional analysis of *CDKN1A* (p21) in senescence (new Appendix Fig S4A and B). Our new data confirm that Day 7 post irradiation U2-OS cells show increased expression of cell cycle inhibitors, and a significant drop in proliferation, evidenced by drop in BrdU uptake.

The authors have checked for the expression of the senescence markers SA- β -Gal and p21 (CDKN1A) in U2-OS cells (Fig S1A). This is not sufficient to prove that the cells are senescent. The senescence state of the cells should be further characterized by checking for more markers of senescence. For example, the expression levels of CDK inhibitors such as p15 (CDKN2B), p16 (CDKN2A) and p53 can be evaluated in both protein and RNA levels. It is not clear at all if U2-OS cells are arrested. In order to verify the cell cycle arrest in the cells, BrdU incorporation in U2-OS cells can be performed.

We have analyzed p53 activity by SDS-PAGE western blotting and show that p53 is phosphorylated on serine 15, indicating ongoing DNA damage response (new Appendix Fig S1E). We have additionally determined that p53 is phosphorylated at serine 46 and serine 391 selectively at later time points using a kinase array (new Fig 4D).

We have performed BrdU incorporation to show that cells are in cell cycle arrest at day 7 post irradiation (new Appendix Fig S1C). Unlike irradiated cells, untreated cells are rapidly proliferating (as is expected for the osteosarcoma cell line).

In summary, our new data further confirm that at day 7 post irradiation U2-OS enter cellular senescence and display an ongoing DNA damage response (using immunofluorescence, SDS-PAGE western blot, GAGE analysis of RNAseq data), constitutive SASP (using RNAseq, RT-qPCR, and cytokine array), and cell cycle arrest (using BrdU incorporation, protein markers of cell cycle arrest, and RT-qPCR, and quantification of cell duplication). We therefore hope that the referee finds our conclusion satisfactory and substantiated that upon irradiated U2-OS cells enter cellular senescence, as manifested by cell cycle arrest and functional SASP.

2. The mouse tissues were not studied in sufficient detail to allow to reach the conclusions authors suggest

a. In Fig 1D murine tissues were irradiated and analyzed for expression of NFKBIA and IL-6 in both skin and kidney tissues. The authors mention in the text a strong activation of NF- κ B, however the results are not shown. This data is critical for verification of the results shown.

On revision of the manuscript, we realized that the term “data not shown” cannot be used for EMBO J articles. Therefore, we took out these phrases in the manuscript, including the statement about kidney and skin tissues, which were referring to our unpublished *in vivo* imaging of irradiated mice. Instead, we cite published literature (p6), which showed that IR strongly activates NF- κ B in various tissues.

b. The in-vivo system should be characterized in terms of senescence and activation of DNA

damage. Relevant markers could be genes that are part of the ds-DDR pathway, for example, p-ATM, p-CHK2, p-p53 and γ H2AX.

We thank the referee for this suggestion. We have used the p53 serine-15 antibody to characterize the DNA damage response *in vivo*. In addition, we have repeated the *in vivo* experiment with 18 mice (n = 6 per group). We stained paraffin sections with anti-p53 serine-15 antibody and quantitated the result (new Appendix Fig S3D and S3E). Our new data show that irradiation also leads to ongoing DNA damage response *in vivo*. Although the majority of cells showing p53 foci in the nucleus, these are cleared by day 7 (or DNA damage is resolved), but a significant number of cells persist showing DNA damage at day 7. We did not detect p53 serine 15 staining in the gut of *IkBa^{IEC.KO}* (*villin-Cre x floxed IkBa*) mice. One could argue that due to hyperproliferation of stem cells in the crypts of these mice (Mikuda et al. 2020), an increase in accumulation of cells with DNA damage should be detected, bearing phosphorylated p53 in the nucleus. But this was not the case. These data imply that NF- κ B is not activated through an active DNA damage response at this late time, but rather due to absence of the inhibitor.

c. The accumulation of senescent cells in the in-vivo system can be done by checking for senescence markers such as p15, p16 and p21 in both protein and RNA levels. The SA- β -Gal staining will also be helpful for this purpose.

We were unfortunately not able to find antibodies for p16 or p21, which would stain paraffin sections in a specific manner. Furthermore, secretory cells (for example Paneth cells) have increased lysozyme content and therefore would stain positive in the absence of senescence. Similarly, quiescent stem cells are p16 positive. This makes it difficult to properly identify senescent cells *in vivo* by staining. For bulk analyses by western blot or RT-PCR, the challenge lies in dilution. Only a few cells are expected to be senescent and therefore only if a target is upregulated very strongly (as is true for SASP in senescence) does it stand a chance to be detected, assuming that only some cells upregulate it.

p16 and p21 are upregulated only 2- to 4-fold. It is therefore unlikely that bulk analysis would be successful for their detection using the treatment conditions we have. Indeed, even though we detected upregulation of select SASP targets during the two phases (new Figs. 1D and EV1H), we did not detect a significant increase in *Cdkn1a*. For this analysis, single cell sequencing might be able to identify cells, which are *Cdkn1a* or *Cdkn2a* positive. This is, however, beyond the scope of the project.

The aim of our *in vivo* investigations was to confirm that two distinct phases of NF- κ B also occur *in vivo*, where the second phase constitutes SASP. However, as discussed in the manuscript, second phase NF- κ B activity does not necessarily have to coincide with senescence.

Indeed, in gut of *IkBa^{IEC.KO}* (*villin-Cre x floxed IkBa*) mice with constitutive NF- κ B, we detected an increase in proliferation, evident by increased number of Ki67+ cells. We have characterized this mouse in our recent paper (Mikuda et al, 2020). Similarly, microarray analysis of RNA from n = 4 mice from each group yielded GO terms showing enrichment of cell cycle or E2F targets. Given that we observed increased proliferation, it is unlikely that constitutive NF- κ B in the mouse gut leads to senescence associated proliferative arrest. We hope the referee finds our answer satisfactory.

d. Importantly, the statistical test used for analyzed the data is incorrect, for the authors compare between 3 groups overall and are using t-test. The more appropriate statistical text for this kind of results is to compare all 3 groups using one-way ANOVA test.

We have now re-analyzed groups of 3 or more using one way ANOVA. We attach EMBOJ-2019-104296_ForRefereesInspection_Statistics where we describe ANOVA test used and distribution.

e. Tissues of I κ B α Δ N mice (Fig 1E) require characterization of DDR, senescence and first phase markers, as suggested above for wt mice.

We have now analyzed kidney cells of I κ B α Δ N mice for expression of *Cdkn1a* and *Cdkn2a* mRNA and show that suppression of NF- κ B does not affect expression of these transcripts (new Appendix Fig 1I).

f. Similarly, tissues of villin-Cre x floxed mice were not studied to the sufficient details that allow to reach the drawn conclusions. The characterization of these samples as suggested above could help to resolve this shortcoming.

We thank the referee for this suggestion. We have recently characterized the I κ B α ^{IEC.KO} (*villin-Cre x floxed I κ B α*) mice and demonstrated that constitutively active p65 not only does not contribute to cell cycle arrest, but on the contrary leads to hyperproliferation of stem-cells and a net increase in Wnt signaling (Mikuda et al, 2020). We also show that constitutive NF- κ B-p65 is not phosphorylated on its IKK substrate site (serine 536). Here, we re-analyzed microarray data and additionally show that transcripts of I κ B α ^{IEC.KO} (*villin-Cre x floxed I κ B α*) mice show an enrichment for a SASP gene signature, in the absence of enrichment for a senescence gene signature (new Fig 2E, new Appendix FigS3C, and Source Data Files for the Figures 2E and S3C). This again indicates that constitutive NF- κ B in epithelial cells drives SASP expression but not senescence associated proliferative arrest.

3. In Fig 3 knockdown should be verified and senescence should be better characterized:

In Fig3A: This figure shows the RNA-seq analysis of the expression levels of SASP targets of NF- κ B following knocking down of p65. The authors should complement this results with a western blot showing the knockdown of p65 in all the relevant experimental groups.

We have characterized the senescence cells for cell cycle arrest by additionally performing BrdU incorporation. We show that knockdown of p65 (shown in Appendix S4A) does not affect cell cycle arrest and does not permit senescent cells to escape from cell cycle arrest (new Appendix Fig S4C).

In addition, we have now included western blot data showing the knockdown efficiency of p65 and levels of p21. Our data confirm that p21 levels are not dependent on p65 (new Appendix Fig S4A).

In Fig3B: This figure shows cell proliferation following knockdown of p65 and irradiation. Cell proliferation by itself is not sufficient to conclude about the senescence state of the cells. The authors should check for different senescence markers (as suggested above in 1).

Furthermore, cell proliferation could be complemented with BrdU incorporation assay, to show cell division.

Using the BrdU assay, we demonstrate that the *RELA* knockdown (new Appendix Fig S4A) has no effect on the proliferative arrest (new Appendix Fig S4C). We firstly showed that U2-OS stop proliferating, as evidenced by lack of BrdU uptake – indeed senescent cells overlapped with negative control (without BrdU) (new Appendix Fig S1C). We then analyzed U2-OS cells bearing shRNA against *RELA*. As is seen before, irradiated cells, regardless of their *RELA* status, show a complete population shift and a dramatic drop in BrdU uptake.

This implies that lack of p65 in senescence does not rescue cells from cell cycle arrest. Since we showed that p65 is the principal subunit both during the first and the second phase of NF- κ B activation (new Fig 3A), we examined the effect of *RELA* knockdown on *CDKN1A* and *CDKN2A*, as suggested by the referee (new Appendix Fig S4A and B). Our new data further support our hypothesis that p65 regulates SASP but not cell cycle arrest in senescence.

4. Fig S1C: Quantification of the staining of both p65 and γ H2AX in all the groups would help to straighten the conclusion.

We appreciate this suggestion and have quantitated nuclear p65 and γ H2AX in all the groups. Our results show significantly increased levels of p65 as a biphasic phenomenon. However, γ H2AX foci appear at an early time point and persist into senescence (Appendix Fig S1D, new right panels).

In order to evaluate the DDR in the cells, the authors could also check for more markers of the ds-DDR pathway, for example, p-ATM, p-CHK2, p-p53 and γ H2AX.

We have performed nuclear/cytoplasmic fractionation of U2-OS cells and have analyzed protein expression of ATM (serine 1981), CHK2 (threonine 68), p53 (serine 15). Analysis by SDS-PAGE western blot shows that, whereas ATM and CHK2 phosphorylation peaks within minutes/hours of DNA damage and then levels off, p53 phosphorylation and presence of γ H2AX foci in the nucleus persist into senescence (new Appendix Fig S1E, S1D, S2F). We corroborated these observations by performing a kinase array analysis with nuclear fractions. Interestingly, whereas p53 serine 15 levels were present at all points following DNA damage, levels of p53 serine 46 and 391 increased steadily and were highest at day 7 (new Fig 4D).

The observation that acute activation of ATM is present within hours and minutes of DNA damage but not in senescence (coinciding with second phase of NF- κ B) and that its activation is not dependent on NF- κ B, is further corroborated by GAGE analysis (See EMBOJ-2019-104296_ForRefereesInspection_R1-3 Fig R2). We have analyzed transcriptional signatures of cells at 1.5h and 7 days post irradiation, and also of cells that have not undergone DNA damage but display constitutive NF- κ B activity due to knockdown of *NFKBIA*. We therefore propose that although DNA damage is the original trigger of both phases of NF- κ B, and that DNA damage response persists (albeit modified) into day 7 (Appendix Fig S1D, S1E and R2), it is not the ATM-IKK DNA damage response axis that activates the second phase of NF- κ B in senescence, but rather suppression of *NFKBIA*.

5. Fig S2A-D: For both BJ and WI-38 cells the authors show in S1D and S1E respectively the levels of p65 following irradiation in different time points. However, in S2B and S2D, the authors use HEK293 and HCT cells respectively to measure I κ B α levels, but they do not show the changes in the levels of p65 to verify the functionality of this systems. Therefore, the authors should include those results.

We have taken out the panels describing HEK293 and HCT cells. Please see our response to the suggestion of Referee 1, minor point 3.

Minor comments:

1. In the paper the authors use various primers from both human and mouse origin in order to check the changes in expression of different genes. However, the authors do not list the primer sequences that were used for the various experiments. Please add all the relevant sequences to the "materials and methods" section.

The complete primer list has now been added, showing primer name, origin and sequence in supplemental methods.

2. The sex of the mice used in the experiments should be mentioned in the text and figure legend. Did the authors performed the experiments on both female and male mice? Is there a significant difference between the sexes?

No difference was observed between sexes, however to avoid age related changes (which are expected to contribute to senescence) we have used young mice <16 weeks of age. For experiments with model animals or irradiation, we used aged matched or siblings with controls and female mice. Selection of animals are further detailed in attached ARRIVE guidelines.

3. Fig S1C: Quantification of the staining of both p65 and γ H2AX in all the groups would help to straighten the conclusion.

In order to evaluate the DDR in the cells, the authors could also check for more markers of the ds-DDR pathway, for example, p-ATM, p-CHK2, p-p53 and γ H2AX.

Please see our response outlined above concerning the p65/ γ H2AX and the DDR marker issues (Main point 4).

4. Fig S2A-D: For both BJ and WI-38 cells the authors show in S1D and S1E respectively the levels of p65 following irradiation in different time points. However, in S2B and S2D, the authors use HEK293 and HCT cells respectively to measure I κ B α levels, but they do not show the changes in the levels of p65 to verify the functionality of this systems. Therefore, the authors should include those results.

We have removed this panel, as described above. Please also see our response to the suggestion of Referee 1, minor point 3.

5. In Fig 1E the expression levels of IL-6 seems to be significantly lower in Δ N mice, in "1.5h" and "6D" time points, compared to the " Δ N untreated" mice. Furthermore, the expression levels of p16 seems to be significantly higher in the " Δ N 6D" group. Can the authors explain these two phenomena?

We have repeated the experiment with n = 4 mice per group. Δ N mice do indeed show increased variability of expression within the group, possibly attributable to their general unhealthy phenotype and therefore predisposition to aberrant stress responses. Note that the previous Fig 1E is now replaced by new Appendix Fig S1I in the modified version.

6. In Fig 2B the authors performed a knockdown I κ B α and check for the changes in NF- κ B over time. The efficiency of the knockdown should be evaluated and presented.

We apologize for this omission and have included the blot showing the efficient knockdown (new Appendix Fig S2F). Please note that in lane 4, I κ B α expression is already down without shRNA induction. This is expected because as we have shown above, in senescent cells, I κ B α is lost.

7. Fig 4C-D: Figure legends are displaced and appear in the legend of figure 5.

We have now corrected this and apologize for the mistake.

8. FigS1B: The amount of irradiation presented on the figure legend (10 Gy) and on the graph itself (20 Gy) are different.

We have corrected this in all the figures.

9. FigS1C: In the figure legend the authors indicate that white arrows point to high nuclear p65 levels. Such arrows are missing from the figure itself.

We have now removed this from the legend, because nuclear p65 is visible without arrows, and its nuclear abundance has now been quantified (see new right panels). Previous Fig S1C is now Appendix Fig S1D.

10. Fig S2A-D: In each cell line the authors use different housekeeping gene for normalization. These genes are not usually used as housekeeping genes. The authors should try to normalize the western blots using one or two classical housekeeping genes, for example, b-Actin/GAPDH.

We have exchanged blots to show standard housekeeping proteins (Appendix Fig S2A and S2B).

11. The authors performed a knockdown ATM and check for the changes in IL-6 and IL-1 α over time - Fig S4B. The efficiency of the knockdown should be evaluated.

Thank you also for this suggestion. We have now included a blot showing the ATM knockdown (new Appendix Fig S5A).

12. The levels of the housekeeping gene used in the "nuclear" part of the figure S4C are not constant between the different lanes and seems to be changing in the same trend as p65. The authors should consider to choose a different housekeeping gene, or to repeat the experiment.

We have exchanged the housekeeping protein to PARP1, which is more appropriate for the nuclear fraction (new Appendix Fig S5C).

Referee #3:

The manuscript by Kolesnichenko and colleagues describes a novel mechanism of NF- κ B activation following DNA damage. Their data shows that a delayed second wave of NF- κ B activation promotes a senescence associated secretome characterised by cytokines and chemokines. The authors demonstrate that expression of NFKBIA which encodes for the NF- κ B inhibitor I κ B is lost following DNA damage and facilitates a subsequent delayed NF- κ B activation wave. The authors propose that the phosphorylation of the RelA subunit at S468 by GSK3 β promotes the transcriptional silencing of NFKBIA following DNA damage. Moreover, the authors propose that the second wave of NF- κ B activity leading to the senescence associated secretory phenotype is independent of the proteasome and IKK kinases. The data presented is of high quality and includes the use of a number of genetic models, in vitro and in vivo. Overall the authors conclusions are justified but the proposed mechanism leading to the regulation of NF- κ B following DNA damage is not sufficiently supported by the data presented.

We thank the referee for this kind evaluation. We have experimentally addressed all the suggestions regarding the mechanism and they are now included in the manuscript.

Major points

1. *Although the authors have focused on the RelA subunit of NF- κ B in this study, the data shown in Figure 1 B would indicate that the cRel subunit is significantly increased during the second wave of activation compared to the first. In addition the EMSA data shown in figure 2b suggests that the DNA binding complexes may be different in the second wave. Have the authors looked at the potential role for cRel or other subunits? If not can they speculate on the potential relevance of these data?*

We thank the referee for bringing up this point. c-Rel showed increased stimulus-induced nuclear localization during both phases, which is expected, given its known cytoplasmic retention by I κ B α . However, c-Rel did not reveal a significant contribution to the global nuclear NF- κ B DNA binding activity, as judged by EMSA super shift analysis, which covered all five NF- κ B family members (new Fig 3A). The latter confirms a predominant role of p65 and p50 containing heterodimers. Of note, the p65 antibody shifted the NF- κ B complex in Fig 3A completely (!), thus all actively DNA binding complexes contain p65. Importantly, we have also shown that knockdown of p65 alone was sufficient to deplete the expression of most NF- κ B dependent SASP targets to basal levels (Fig 3B). Although we cannot exclude that c-Rel regulates gene transcription in senescence, at least for the SASP genes analyzed, p65 is sufficient and necessary for their expression.

2. *The author demonstrate a lack of NFKBIA expression following the DNA damage event. What are the expression patterns for NFKBIB (I κ B β) and NFKBIE (I κ B ϵ)? Are they detected in the RNA-seq data and do they follow a similar pattern to that of I κ B α ? Can the authors rule out a role for these related proteins?*

We have analyzed expression of NFKBIB (I κ B β) and did not detect a significant change (Fig R3). Its expression at the mRNA level and also that of I κ B ϵ was likewise not significantly altered under the conditions tested (EV Table 1A).

3. *The data in Figure 3 indicates that RelA promotes the expression of the SASP gene set but does not promote cell cycle arrest. The authors should measure the expression of known regulators of senescence associate cell cycle arrest such as CDKN1A and CDKN2A in control and RelA knockdown cells to demonstrate whether the lack of effect on cell cycle arrest may be related to the expression of such factors.*

We thank the referee for this helpful suggestion. To further characterize the contribution of p65 to SASP versus cell cycle arrest, we added a p21 western blot (Appendix Fig S4A) and BrdU analyses (new Appendix Fig S1C, S3B, S4C). Both show that p65 does not regulate senescence associated cell cycle arrest. Analyses of NF- κ B activity corroborated these findings - we showed that *in vivo*, NF- κ B promotes hyperproliferation of stem cells in intestinal crypts (Mikuda et al., 2020). Suppression of NF- κ B through the super-repressor I κ B α Δ N suppresses expression of SASP factors, but does not alter expression of *Cdkn1a* or *Cdkn2a* (Appendix Fig S1I).

4. *The data presented on the role of phosphorylation of p65 at S 468 is not very convincing and doesn't strongly support the authors proposal that phosphorylation at S468 controls of NFKB1A expression. The western blot data shown in figure 4 B should be quantified and I κ B α expression normalised to levels of overexpressed p65 to more clearly demonstrate that the S468A mutation increases the expression of I κ B α . The authors should also probe these samples with an antibody for pS468 and S267. Similarly the data shown in figure 4C and 4D*

should also be quantified and the level of IκBα normalised to the levels of p65 pS468. For these two experiments the level of total p65 should also be measured. The analysis of the effects of 10mM LiCl treatment on p65 phosphorylation needs to be controlled by cells treated with an equivalent concentration of NaCl or similar salt.

We thank the referee for bringing up this point. We have quantitated the levels of IκBα and p65 and present data in new Table R2. We would like to point out that it is not the presence of phosphorylation at serine 468, but rather absence that permits expression of *NFKBIA*.

To further substantiate our conclusion, and characterize the role of the S468 phosphorylation site, we have performed ChIP analysis with total p65 and phospho-serine 468 antibodies. In senescence but not during the first phase of NF-κB activation, the *NFKBIA* locus was inducibly bound by inhibitory p65-pSer468. We therefore propose the following model: in senescence, phosphorylated p65 (pSer468) binds to *NFKBIA* gene, however ectopic overexpression of p65 (not phosphorylated at the inhibitory site S468) out titrates phosphorylated p65 and thus permits *NFKBIA* to be transcribed. In accordance with this, overexpressed p65 plasmid is not phosphorylated at S468 (Fig R1).

We did not detect a difference in the p65 phosphorylation state when using either equivalent NaCl, water, or DMSO. However, to additionally substantiate our conclusion regarding the involvement of GSK, we used a more specific inhibitor, CHIR99021. Inhibition of GSK in senescence partially rescues *NFKBIA* expression, and leads to concomitant suppression of *IL6* expression (new Fig 4G).

5. The authors use proteasome inhibitors and deletion of IKKβ and NEMO to show that the second wave of NF-κB activity is independent of IKK and the proteasome. However these data do not rule out the possible involvement of IKKα and do not justify the authors conclusion. The authors need to rule out the possible role of IKKα in order to come to the conclusion that the activity is IKK-independent.

We are grateful for this suggestion which allowed us to reach completeness in our analysis of IKK activity during the second phase of NF-κB activation. We have performed an IKKα (encoded by *CHUK*) knockdown and show that IKKα does not regulate SASP by checking *bona fide* SASP targets by RT-qPCR (new Appendix Fig S5D). This is also consistent with the absence of p52 and RelB in the compositional subunit-analysis (new Fig 3A). The efficient IKKα knockdown is seen in the *CHUK* RT-qPCR sample (new Appendix Fig S5D). We show that most SASP targets tested, which were IKKβ independent, were also IKKα independent. Interestingly, the induction of *IL6* and *IL1A* expression levels were partially reduced by IKKα knockdown. This might be due to yet poorly understood gene-specific nuclear functions of IKKα at the chromatin level, which are not related to NF-κB activation (reviewed in Hinz and Scheidereit, 2014, EMBO Reports).

In summary, our new findings continue to support our hypothesis that the second phase of NF-κB, as reflected by NF-κB-regulated SASP genes, is essentially IKK independent.

Minor points:

1. The RNA-seq data should be made- available in open access database such as EMBL ArrayExpress or NCBI GEO as per the journal guidelines.

We have now made RNA-seq data available at NCBI GEO with accession number GSE 158743. For animal experiments we are attaching ARRIVE guidelines. All primer sequences are added into supplemental information.

2. For most experiments shown the figure legends do not indicate how many independent replicate experiments were performed.

We thank the referee for noting this point. We have now indicated the number of biological replicates, and technical repeats. In addition, we have included ARRIVE guidelines to more extensively describe mouse experiments and indicate the numbers used. We have also included statistical analyses for experiments and show statistical significance in the figure and the legend. Please also see our extensive response to referee 2 concerning statistics and the summary in EMBOJ-2019-104296_ForRefereesInspection_Statistics.

3. The figure legend for figure 4 parts C and D appears on the legend for figure 5.

We apologize for this misplacement, which was now corrected.

Figure R1

SDS-PAGE western blot of nuclear and cytoplasmic lysates (as in Fig 4B), using an anti p65 pSer468 antibody. Please note the different loading order compared to Fig 4B. Lanes 3-10 show lysates from cells treated with Dox to knock down endogenous p65.

Figure R2

GAGE analysis performed on RNAseq data as described previously. Enrichment of selected gene sets with vertical black line representing p value cut off at 0.05.

Figure R3

SDS-PAGE western blot on whole cell lysates from U2-OS cells, irradiated at time points indicated 20 Gy.

Thank you for submitting your revised manuscript for our editorial consideration, and please excuse the delay associated with its re-evaluation. All three original referees have now reviewed it again, and given their positive comments (see below), we shall be happy to accept your manuscript for EMBO Journal publication, after incorporation of a number of editorial points as follows:

REFEREE REPORTS

Referee #1:

The authors have conscientiously responded to all critiques. This is a valuable study which addresses a poorly understood topic, regulation of NFkB in senescent cells. While there are unanswered questions, this MS contains important data for the field. I recommend publication in EMBO J.

Referee #2:

The authors addressed the comments in an adequate manner. I have no further comments.

Referee #3:

The authors have fully addressed the issues raised in my initial review and the updated manuscript includes additional experimental data to support the authors conclusions.

The authors have made all requested editorial changes.

Accepted

9th Dec 2020

Thank you for submitting your final revised manuscript for our consideration. I am pleased to inform you that we have now accepted it for publication in The EMBO Journal.

Scheidereit, Claus

EMBO J

Manuscript Number: EMBOJ-2019-104296